# NECAPs are negative regulators of the AP2 clathrin adaptor complex

Gwendolyn M Beacham[1], Edward A Partlow[1], Jeffrey J Lange[2], Gunther Hollopeter[1]*

[1]Department of Molecular Medicine, Cornell University, Ithaca, United States; [2]Stowers Institute for Medical Research, Kansas City, United States

**Abstract** Eukaryotic cells internalize transmembrane receptors via clathrin-mediated endocytosis, but it remains unclear how the machinery underpinning this process is regulated. We recently discovered that membrane-associated muniscin proteins such as FCHo and SGIP initiate endocytosis by converting the AP2 clathrin adaptor complex to an open, active conformation that is then phosphorylated (Hollopeter et al., 2014). Here we report that loss of *ncap-1*, the sole *C. elegans* gene encoding an adaptiN Ear-binding Coat-Associated Protein (NECAP), bypasses the requirement for FCHO-1. Biochemical analyses reveal AP2 accumulates in an open, phosphorylated state in *ncap-1* mutant worms, suggesting NECAPs promote the closed, inactive conformation of AP2. Consistent with this model, NECAPs preferentially bind open and phosphorylated forms of AP2 in vitro and localize with constitutively open AP2 mutants in vivo. NECAPs do not associate with phosphorylation-defective AP2 mutants, implying that phosphorylation precedes NECAP recruitment. We propose NECAPs function late in endocytosis to inactivate AP2.

DOI: https://doi.org/10.7554/eLife.32242.001

*For correspondence:
gh383@cornell.edu

Competing interests: The authors declare that no competing interests exist.

## Introduction

Eukaryotic cells internalize transmembrane protein cargo, such as laden receptors, by enrobing cargo-containing regions of the plasma membrane with a cytosolic clathrin coat. The GTPase dynamin releases the nascent transport vesicle into the cytosol for subsequent delivery to internal organelles. This fundamental cellular process is called clathrin-mediated endocytosis. The Adaptor Protein 2 (AP2) complex, a heterotetramer of α, β2, μ2, and σ2 subunits (*Figure 1—figure supplement 1A*), actively couples polymerization of the clathrin coat to membrane phospholipids and cargo molecules destined for internalization (*Kirchhausen et al., 2014*). In this manner, AP2 choreographs turnover of the cell surface and entry into the endolysosomal system, yet it remains unclear how this complex is regulated with spatiotemporal precision to maintain appropriate levels of endocytosis.

AP2 activity is likely regulated at the structural level. Biochemical (*Matsui and Kirchhausen, 1990*; *Rapoport et al., 1997*) and structural data suggest that AP2 adopts at least two functionally distinct conformations (*Figure 1—figure supplement 1B*). In one arrangement, the binding pockets for membrane, cargo, and clathrin are occluded; this orientation is thought to represent a closed, inactive state (*Collins et al., 2002*). Molecular rearrangement of the AP2 subunits exposes these binding sites and results in an open, active complex that presumably coordinates the formation of endocytic pits (*Jackson et al., 2010*; *Kelly et al., 2014*; *Kelly et al., 2008*).

Multiple inputs coordinate the conformational rearrangement of AP2 but recent data suggest that membrane-associated muniscin proteins (*Reider et al., 2009*) allosterically activate the AP2 complex (*Hollopeter et al., 2014*; *Umasankar et al., 2014*). These include SH3-containing GRB2-like protein 3-Interacting Protein (SGIP) (*Uezu et al., 2007*) and Fer/CIP4 Homology domain only (FCHo) proteins (*Henne et al., 2010*; *Sakaushi et al., 2007*). In *C. elegans*, loss of the sole muniscin, FCHO-1, causes AP2 to dwell in the inactive state, leading to hindered endocytosis

(*Hollopeter et al., 2014*). Residual endocytosis in *fcho-1* mutants is likely coordinated by additional endocytic components acting in parallel (*Ma et al., 2016*; *Mayers et al., 2013*; *Wang et al., 2016*). Gain-of-function mutations in AP2 that selectively destabilize the closed conformation, thereby artificially inducing the open state, can bypass loss of FCHO-1. A conserved region of muniscins called the AP2 activator domain (APA), binds AP2 and is also sufficient to rescue both *fcho-1* mutants (*Hollopeter et al., 2014*) and FCHo-deficient tissue culture cells (*Umasankar et al., 2014*).

Activation of AP2 and subsequent clathrin binding also require additional molecular events, including interactions with PI(4,5)P$_2$ and cargo (*Ehrlich et al., 2004*; *Jackson et al., 2010*; *Kadlecova et al., 2017*; *Kelly et al., 2014*; *Kelly et al., 2008*). Phosphorylation of the AP2 μ2 subunit by an AP2-Associated Kinase (AAK1) (*Conner and Schmid, 2002*) is also proposed to be required for endocytosis (*Olusanya et al., 2001*; *Ricotta et al., 2002*) and is associated with the open form of AP2 (*Hollopeter et al., 2014*; *Höning et al., 2005*). However, it is not entirely clear whether phosphorylation induces the open state, stabilizes the open state (*Jackson et al., 2010*; *Kadlecova et al., 2017*), or marks adaptor complexes that have already been incorporated into clathrin coats (*Conner et al., 2003*; *Jackson et al., 2003*; *Semerdjieva et al., 2008*).

How then is AP2 inactivated and returned to the cytosol after endocytosis is complete? The heat shock protein Hsc70 (*Chappell et al., 1986*) and its cofactors (*Greener et al., 2000*; *Umeda et al., 2000*; *Ungewickell et al., 1995*) remove clathrin coats from vesicles. Release of AP2 is also thought to depend on Hsc70 (*Hannan et al., 1998*) but appears to require additional factors that stimulate the dephosphorylation of μ2 (*Ghosh and Kornfeld, 2003*) and the conversion of vesicular PI(4,5)P$_2$ into PI(4)P (*Cremona et al., 1999*; *Semerdjieva et al., 2008*). Whether there are mechanisms to directly restore the inactive, closed conformation of AP2 has been relatively unexplored.

In this study, we identify adaptiN Ear-binding Coat-Associated Proteins (NECAPs) as AP2 modulators that promote inactivation of the complex. NECAPs were originally identified as endocytic accessory proteins through proteomic analysis of clathrin-coated vesicles (*Wasiak et al., 2002*) and were shown to bind the AP2 α appendage via a C-terminal WXXF motif (*Ritter et al., 2004*; *Ritter et al., 2003*). A conserved N-terminal region of these proteins (the PHear domain) appears to be structurally similar to pleckstrin homology domains and exhibits varying affinity for FxDxF motif-containing endocytic accessory proteins (*Ritter et al., 2007*). While most invertebrate and fungal genomes encode a single NECAP, vertebrates express two closely-related forms (*Dergai et al., 2016*; *Manna et al., 2015*) that are thought to be functionally distinct. Brain-enriched NECAP1 is proposed to modulate the size and number of endocytic structures by regulating the binding of clathrin to the AP2 β2 linker and the coordination of accessory protein recruitment to the α appendage (*Ritter et al., 2013*). In support of this model, NECAP1 has been localized to clathrin-coated pits at the ultrastructural level (*Sochacki et al., 2017*). By contrast, ubiquitously-expressed NECAP2 has been proposed to recruit a different clathrin adaptor, AP1, to early endosomes to facilitate fast recycling of receptors back to the cell surface (*Chamberland et al., 2016*).

Using an unbiased genetic screen in *C. elegans*, we have unveiled a novel role for NECAPs as proteins that work in opposition of muniscins. We demonstrate that in worms lacking NECAPs (*ncap-1* mutants), AP2 accumulates in an open, hyper-phosphorylated state and that heterologous NECAPs restore the closed conformation. NECAPs bind open and phosphorylated forms of the AP2 core but not phosphorylation-defective AP2 mutants, suggesting that NECAPs target the active complex. Together, our genetic and biochemical evidence establish NECAPs as negative regulators of AP2.

## Results

### Loss of *ncap-1* suppresses *fcho-1* mutants

By mutagenizing worms lacking *fcho-1* and selecting for offspring that outcompete their siblings (*Hollopeter et al., 2014*), we isolated nine independent loss-of-function mutations in *ncap-1,* which encodes the sole adaptiN Ear-binding Coat-Associated Protein (NECAP) in *C. elegans* (*Figure 1A*). Worms with null mutations in *fcho-1* exhibit reduced fitness and require twice as long as wild type worms to populate a culture plate and consume the bacterial food source. Additionally, they display a distinctive 'jowls' phenotype that is indicative of compromised AP2 activity (*Gu et al., 2013*;

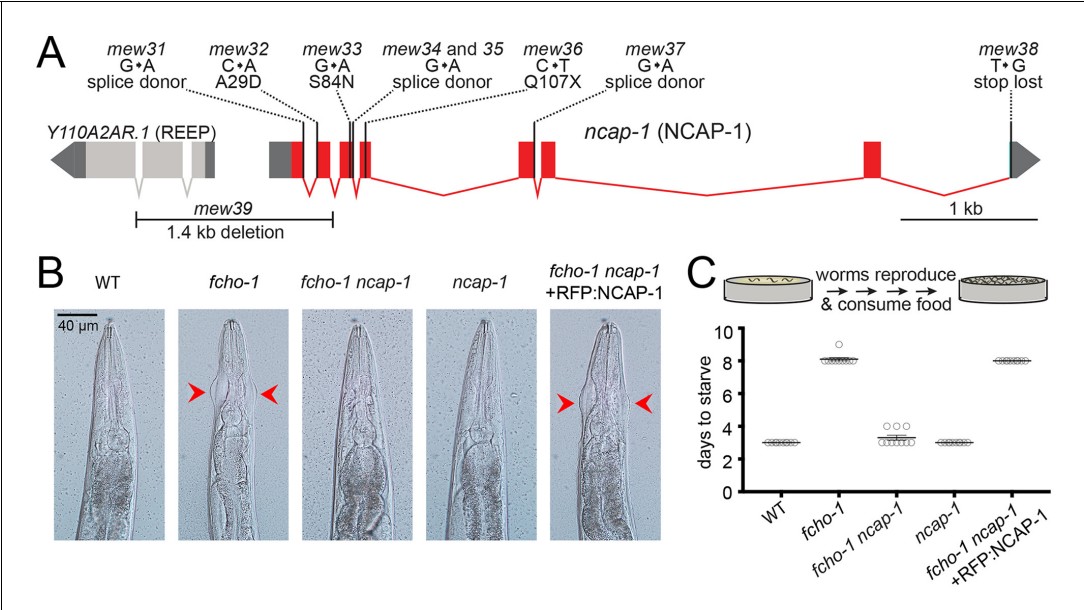

**Figure 1.** Loss of NCAP-1 suppresses *fcho-1* mutants. (**A**) Gene model of the *C. elegans ncap-1* locus. Boxes represent exons. Mutations isolated from the *fcho-1* suppressor screen are indicated. The deletion allele, *mew39*, was used throughout this study as *ncap-1*. The neighboring gene (Y110A2AR.1) is predicted to encode a receptor expression-enhancing protein (REEP). (**B**) Animal heads showing jowls phenotype (red arrows). Anterior is up. WT, wild type; RFP:NCAP-1, red fluorescent protein-tagged NCAP-1 single-copy transgene. (**C**) Starvation assay. Data represent days required for worms to reproduce and consume bacterial food source (top schematic). Bars indicate mean ±SEM for n = 10 biological replicates.

DOI: https://doi.org/10.7554/eLife.32242.002

The following figure supplement is available for figure 1:

**Figure supplement 1.** AP2 structures and mutations.
DOI: https://doi.org/10.7554/eLife.32242.003

*Hollopeter et al., 2014*). Loss of NCAP-1 suppressed the jowls phenotype (*Figure 1B*) and ameliorated the fitness of *fcho-1* mutants (*Figure 1C*). Expression of fluorescently-tagged NCAP-1 in *fcho-1 ncap-1* double mutants restored the *fcho-1* phenotype (*Figure 1B and C*), confirming that suppression of *fcho-1* is due to loss of NCAP-1 function.

Previously we observed that suppression of *fcho-1* correlates with recovery of AP2 activity (*Hollopeter et al., 2014*). To evaluate if loss of NCAP-1 also increases AP2 activity in *fcho-1* mutants, we imaged GFP-tagged AP2 α adaptin (APA-2:GFP) in macrophagic cells called coelomocytes that exhibit robust levels of endocytosis (*Sato et al., 2014*). Muniscins, such as FCHO-1, stabilize AP2 on the plasma membrane to promote its incorporation into presumptive pits, or clusters (*Cocucci et al., 2012*; *Henne et al., 2010*; *Hollopeter et al., 2014*). We performed Fluorescence Recovery After Photobleaching (FRAP) to quantify AP2 stability on the coelomocyte membrane. In the *fcho-1* mutants, fluorescent AP2 recovers approximately three times faster than in wild type worms, indicating that AP2 association with the membrane is destabilized (*Hollopeter et al., 2014*). Loss of NCAP-1 slowed AP2 kinetics in *fcho-1* animals (*Figure 2A*), suggesting that AP2 is incorporated into longer-lived structures. Indeed, we observed improved AP2 clustering in some cells, but the trend was not significant (*Figure 2B*). Endocytosis of an AP2-dependent model cargo is compromised in *fcho-1* mutants (*Hollopeter et al., 2014*) whereas cargo internalization was partially rescued in *fcho-1 ncap-1* worms (*Figure 2C*). These data indicate that a loss of NCAP-1 improves AP2 activity in *fcho-1* mutants.

## Open and phosphorylated AP2 accumulates in *ncap-1* mutants

We previously discovered that AP2 activity in *fcho-1* mutants is also partially restored by amino acid substitutions that specifically destabilize the closed conformation of the adaptor complex, henceforth referred to as 'open AP2 mutations' (*Hollopeter et al., 2014*). Because loss of NCAP-1 phenotypically mimicked open AP2 mutations, we tested whether AP2 also dwells in the open state in

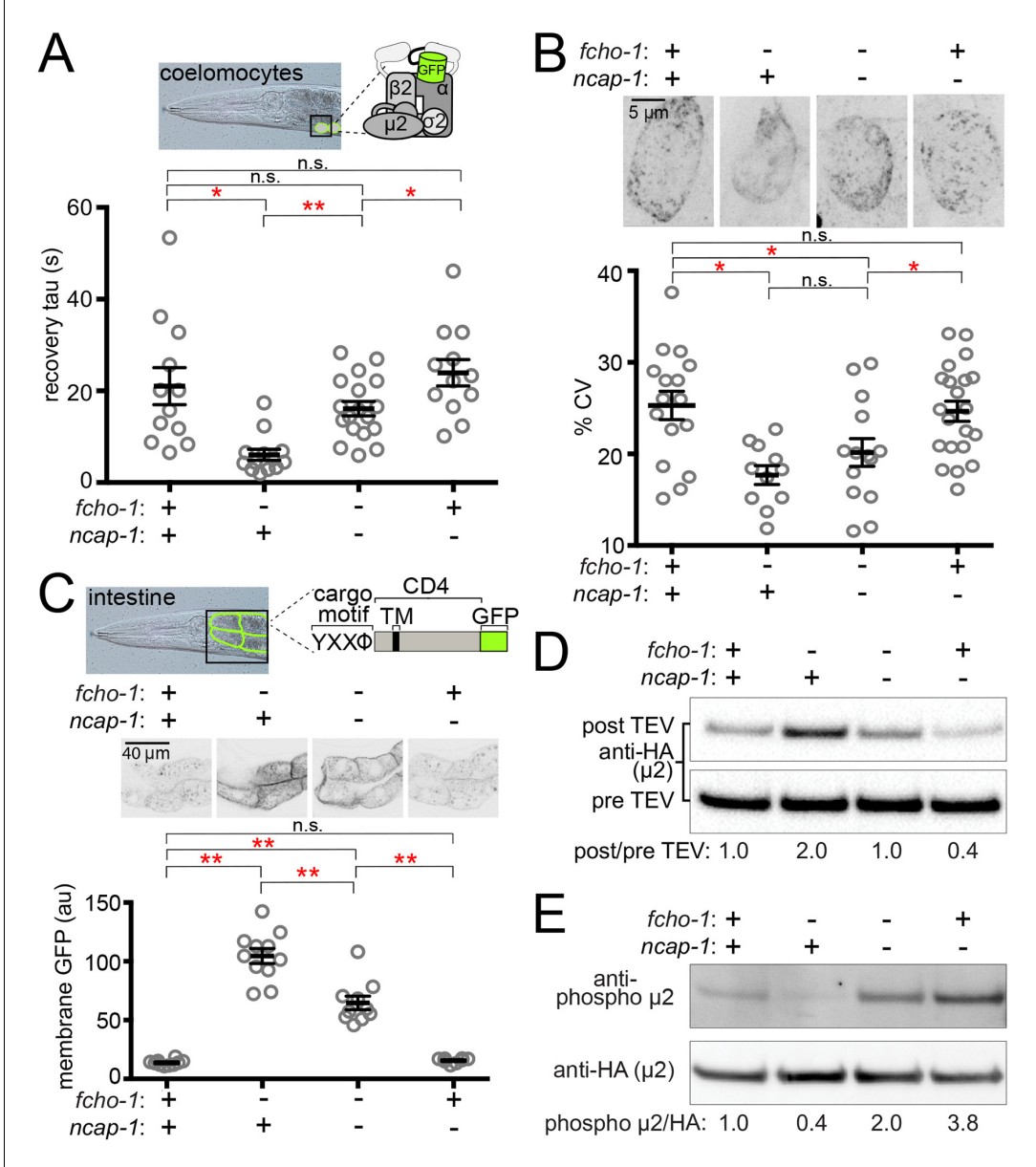

**Figure 2.** Loss of NCAP-1 restores AP2 activity in *fcho-1* mutants. (**A**) FRAP analysis of GFP-tagged AP2 α adaptin (APA-2:GFP) on membranes of coelomocytes (top schematic). Time constants (tau) of the fluorescence recovery are plotted. (**B**) AP2 localization in coelomocytes. Representative confocal images of coelomocytes in worms expressing APA-2:GFP. Micrographs (top) are representative maximum projections of Z-slices through approximately half of a cell. Data represent the coefficient of variance (%CV) of pixel intensities for individual cells. (**C**) Artificial AP2 cargo assay. Representative confocal micrographs of intestinal cells (middle) in worms expressing a GFP-tagged cargo (top schematic). TM, transmembrane domain. The average pixel intensity along a basolateral membrane was measured (bottom). (**A–C**) Bars indicate mean ±SEM for n ≥ 8 biological replicates. *p<0.05, **p<0.001, not significant (n.s.), unpaired, two-tailed T-test. (**D**) μ2 protease-sensitivity assay. Western blot analysis of whole worm lysates was used to quantify the amount of full-length μ2 (anti-HA, 50 kDa) before (pre TEV, bottom blot) and after protease induction (post TEV, top blot). Band intensities were compared to a tubulin loading control and normalized to the *fcho-1*(+) *ncap-1*(+) ratio (values below). (**E**) μ2 phosphorylation assay. Western blot analysis of whole worm lysates to quantify phosphorylated μ2 (top blot) relative to total μ2 subunit (bottom blot). Values indicate band intensity ratios of phospho μ2 compared to total μ2, normalized to the *fcho-1*(+) *ncap-1*(+) ratio (values below). (**D and E**) Blots are representative of ≥3 biological replicates. +, wild type allele; -, deletion allele.

DOI: https://doi.org/10.7554/eLife.32242.004

*fcho-1 ncap-1* worms. We evaluated AP2 conformation using an in vivo protease-sensitivity assay (*Hollopeter et al., 2014*). Briefly, a Tobacco Etch Virus (TEV) protease site was inserted into a surface loop of the μ2 subunit that becomes more exposed when AP2 is opened (*Jackson et al., 2010*; *Matsui and Kirchhausen, 1990*). Following induction of TEV protease expression, the ensemble protease-sensitivity of the AP2 complexes is determined using western blot analysis of μ2 in whole worm lysates. In *fcho-1* worms, AP2 dwells in a protease-insensitive, closed state (*Hollopeter et al., 2014*) and AP2 was more protease-sensitive (open) in *ncap-1* mutants (*Figure 2D*). These results explain the phenotypic rescue of *fcho-1* mutants and indicate that AP2 accumulates in an open state in the absence of NCAP-1.

Phosphorylation of μ2 correlates with open AP2 in *C. elegans* (*Hollopeter et al., 2014*). Because loss of NCAP-1 increases AP2 protease-sensitivity, we wondered whether μ2 phosphorylation might also be elevated. We quantified the phosphorylation status of AP2 in vivo by blotting worm lysates with an antibody specific to the phosphorylated threonine in μ2 (T160). Loss of NCAP-1 resulted in T160 phosphorylation levels greater than in wild type worms (*Figure 2E*). Thus, AP2 adopts a hyper-phosphorylated, open state without the action of NCAP-1.

## Negative regulation of AP2 is a conserved function of NECAPs

To determine if the ability to inactivate AP2 is a conserved function of NECAPs in the context of our *C. elegans* system, we expressed heterologous NECAPs as single-copy transgenes in *fcho-1 ncap-1* mutant worms. Mouse NECAP1 and NECAP2, as well as a NECAP from the multicellular fungus, *Sphaerobolus stellatus,* recapitulated the reduced fitness of *fcho-1* mutants (*Figure 3A*) and caused AP2 to adopt a more closed, protease-insensitive state (*Figure 3B*). It appears that NECAPs have retained the ability to negatively regulate AP2, at least in *C. elegans*.

## NECAPs bind the open and phosphorylated AP2 core in vitro

We investigated the possibility that NECAPs negatively regulate AP2 via a direct interaction. Indeed, affinity-tagged NECAPs co-purified endogenous AP2 complexes from HEK293 cells (*Figure 4A*), consistent with previous reports (*Ritter et al., 2003*). NECAPs are thought to bind the AP2 α adaptin appendage domain and the clathrin binding box in the β2 adaptin hinge region via a C-terminal WXXF motif and an N-terminal PHear domain, respectively (*Ritter et al., 2013*; *Ritter et al., 2003*). Because the C-terminal WXXF motif is poorly conserved in *C. elegans* NCAP-1 (LLDF) and completely absent from the *S. stellatus* protein (PKRR), it was unclear whether the negative regulation of AP2 by heterologous NECAPs (*Figure 3*) was mediated via binding to the α appendage. Instead, we were interested to examine whether NECAPs also bind to the AP2 core (*Figure 1—figure supplement 1B*) because this would offer a direct route to conformational regulation. We purified recombinant vertebrate AP2 cores lacking ears and linkers and tested binding of these cores to recombinant NECAPs from worms and mice (*Figure 4B*). Interestingly, NECAPs did not bind unmodified AP2 cores (*Figure 4B*).

Because open and phosphorylated AP2s accumulate in *ncap-1* mutants, we reasoned that NECAPs might act upon these modified forms of AP2. To test our hypothesis, we introduced a previously-characterized open AP2 mutation in the μ2 subunit (E302K; *Figure 1—figure supplement 1C*) (*Hollopeter et al., 2014*) to produce recombinant AP2 cores that dwell in the open state. We also co-expressed AP2 with the kinase domain of AAK1 to purify AP2 cores with phosphorylated μ2 (T156) (*Höning et al., 2005*). When these modified complexes were tested in our pulldown assays, we observed that NECAPs bound both open and phosphorylated AP2 (*Figure 4B and C*). These data suggest that NECAPs associate with the AP2 complex in a conformation-dependent manner.

## NCAP-1 localizes with constitutively open AP2 in vivo

To corroborate our in vitro pulldown results we sought in vivo evidence that NECAPs associate with open and phosphorylated AP2. In *C. elegans*, fluorescently-tagged AP2 is enriched at the nerve ring, a major neuropil of bundled axons encircling the pharynx (*White et al., 1986*) (*Figure 5*). We used confocal microscopy to observe localization of NCAP-1 tagged with a red fluorescent protein (RFP:NCAP-1) at the nerve ring in live worms. In both wild type and *fcho-1* mutant worms, NCAP-1 was not overtly enriched compared to AP2. The low level of NCAP-1 relative to AP2 may indicate that in worms, NCAP-1 is not stably associated with AP2 on membranes. However, introduction of a

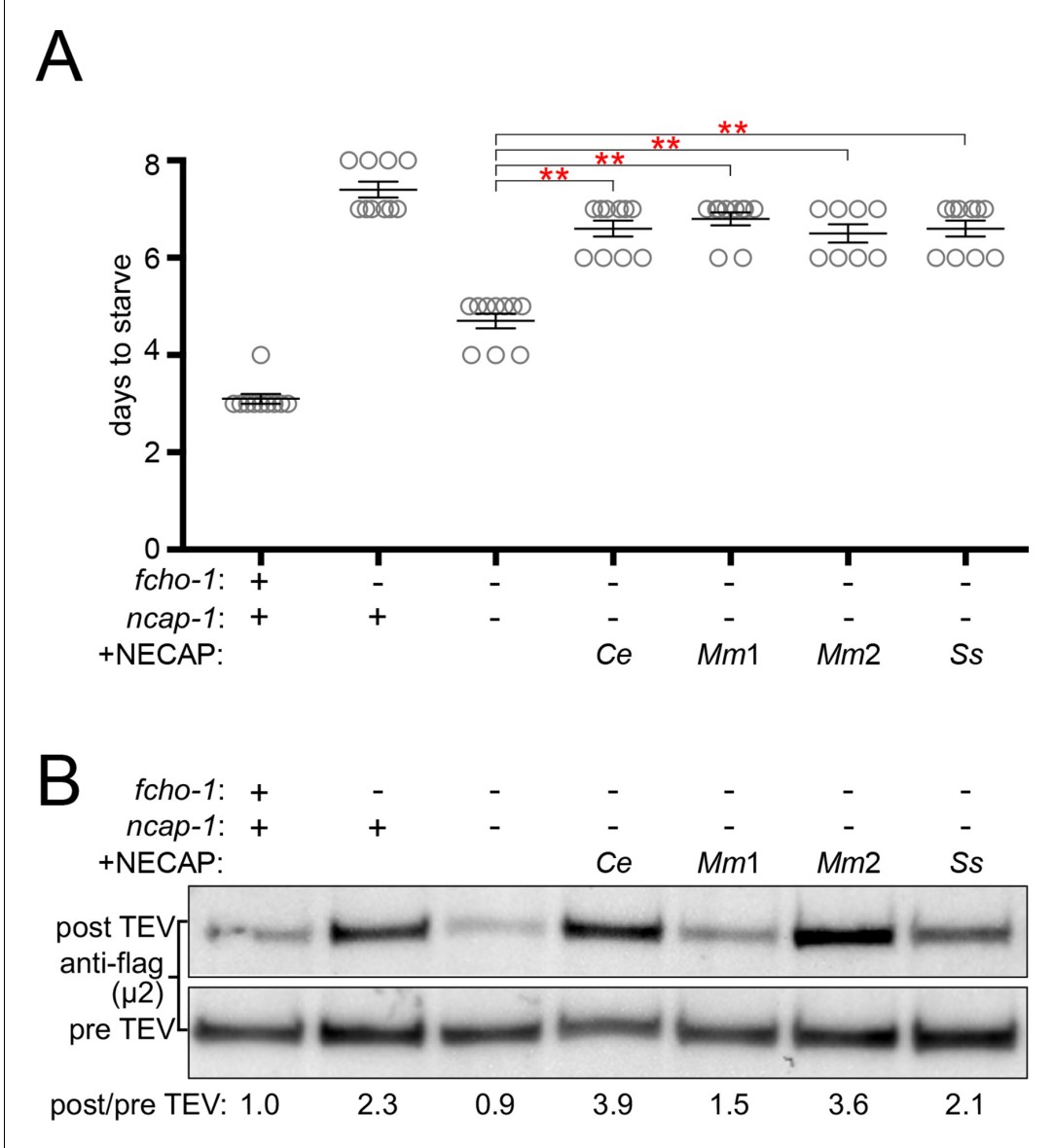

**Figure 3.** NECAPs restore closed AP2 in *fcho-1 ncap-1* worms. RFP-tagged NECAPs were expressed as single copy transgenes in *fcho-1 ncap-1* worms. *Ce, C. elegans; Mm, M. musculus; Ss, Sphaerobolus stellatus* (multicellular fungus). +, wild type allele; -, deletion allele. (**A**) Starvation assay performed as in *Figure 1C*. Bars represent mean ±SEM for n ≥ 7 biological replicates. **p<0.001, unpaired, two-tailed T-test. (**B**) μ2 protease-sensitivity assay as in *Figure 2D*, except a flag-tagged μ2 subunit was used. Band intensities were compared to a histone loading control and normalized to the *fcho-1*(+) *ncap-1*(+) ratio (values below). Blot is representative of 2 biological replicates.

DOI: https://doi.org/10.7554/eLife.32242.005

mutation in μ2 known to generate hyper-phosphorylated, constitutively open AP2 (E306K; *Figure 1— figure supplement 1C*) (*Hollopeter et al., 2014*) enhanced localization of NCAP-1 nearly 2-fold (*Figure 5*). In other words, NCAP-1 was abnormally recruited to the nerve ring in the open AP2 mutant suggesting that NECAPs favor association with activated forms of AP2 in vivo. It is possible that the basal level of localization observed in both wild type and *fcho-1* mutants may represent NCAP-1 bound to the AP2 appendages (*Ritter et al., 2013*; *Ritter et al., 2003*).

## AP2 phosphorylation site mutations weaken NECAP binding

We were curious whether phosphorylation of AP2 is a prerequisite for NECAP binding. Interestingly, in *C. elegans*, mutating the phosphorylation site, μ2(T160), to an alanine, isoleucine, proline, or

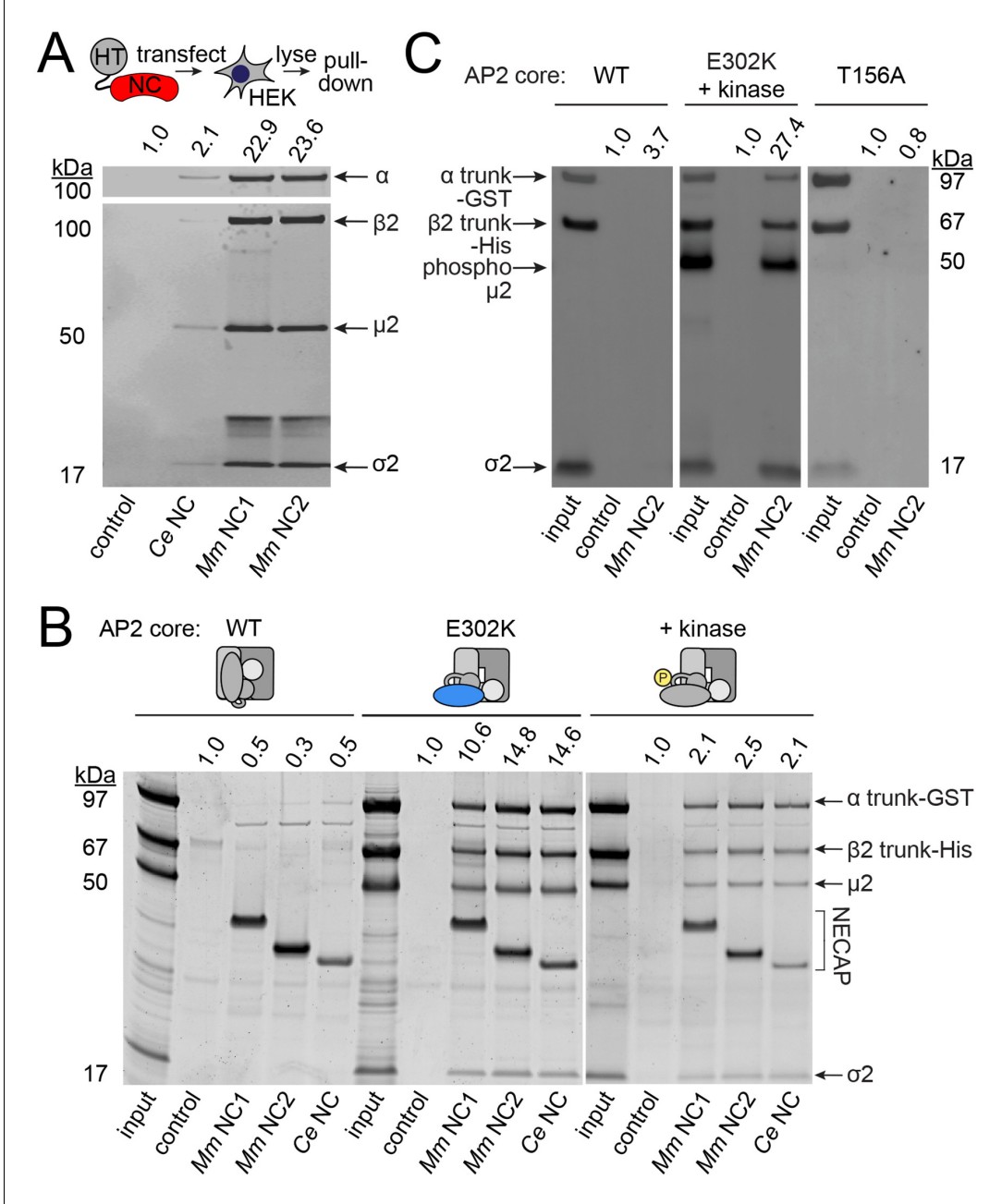

**Figure 4.** NECAPs bind the open and phosphorylated AP2 core. Pulldown assays using affinity-tagged NECAPs. Proteins were cleaved from the affinity tag (HaloTag), electrophoretically separated and then blotted for AP2 subunits (**A** and **C**) or SYPRO-stained prior to imaging (**B**). Control, HaloTag alone; NC, NECAP; *Ce*, *C. elegans; Mm, M. musculus*. (**A**) Western blot analysis (middle) of samples purified from human cell lysates (top schematic) expressing the indicated NECAP bait (bottom). (**B** and **C**) In vitro pulldown assays using purified recombinant bait (NECAPs, bottom) and prey (vertebrate AP2 cores, top). Co-expression with the kinase domain from mouse AAK1 (+kinase) generates phosphorylated AP2. Amino acid changes in µ2 are indicated: E302K, constitutively open AP2; T156A, phosphorylation-defective AP2; see also *Figure 1—figure supplement 1C*. 20% of prey input was analyzed for comparison with 50% of the sample released by the protease. (**A–C**) Band intensities of the α subunit (**A**) or the α trunk (**B** and **C**) were quantified, background signal subtracted, and values normalized to the HaloTag control (values above). Data are representative of 2 biological (**A**), one technical (**B**), and two technical (**C**) replicates.

DOI: https://doi.org/10.7554/eLife.32242.006

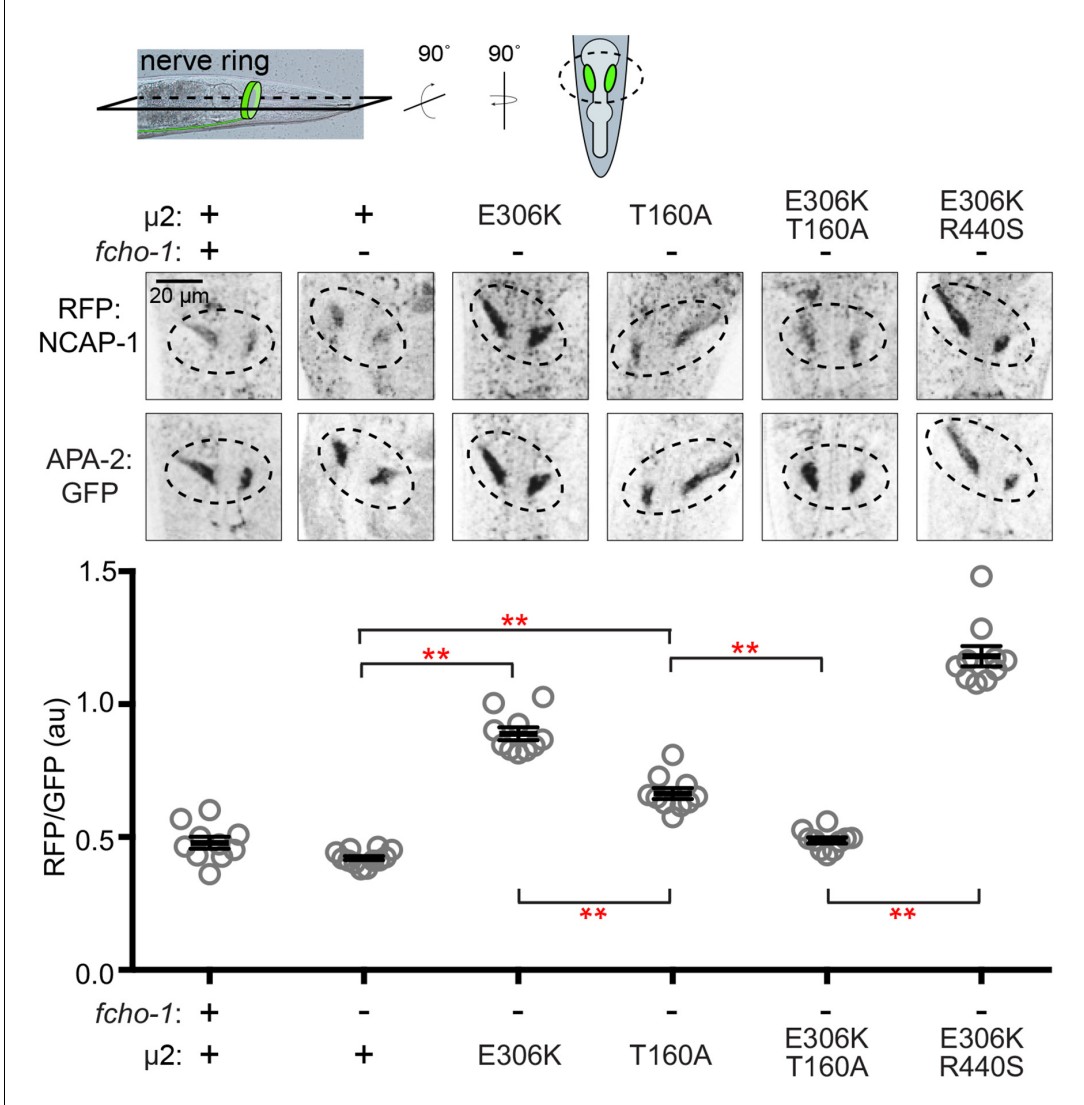

**Figure 5.** Phosphorylated AP2 recruits NCAP-1 in vivo. Representative confocal slices (middle) through the approximate center of the nerve ring of worms (top schematic) expressing RFP:NCAP-1 and APA-2:GFP. RFP to GFP signal intensity at the nerve ring is plotted (bottom). Mutations in μ2 are indicated: E306K and R440S, constitutively open AP2; T160A, phosphorylation-defective AP2; see also *Figure 1—figure supplement 1C*. Bars indicate mean ±SEM for n ≥ 10 biological replicates. **p<0.001, unpaired, two-tailed T-test. au, arbitrary units; +, wild type allele; -, deletion allele.
DOI: https://doi.org/10.7554/eLife.32242.007

glutamate suppresses *fcho-1* and produces an open complex according to the in vivo protease-sensitivity assay (*Hollopeter et al., 2014*). However, when we purified vertebrate AP2 cores containing the phosphorylation site mutation (T156A; *Figure 1—figure supplement 1C*) and tested their ability to bind mouse NECAP2, we observed very little interaction in pulldown assays (*Figure 4C*). This result suggests that NECAPs do not bind phosphorylation-defective AP2. To determine whether the phosphorylation site mutants also impact NCAP-1 association with AP2 in vivo, we examined the localization of RFP:NCAP-1 in μ2(T160A) mutant worms. Compared to mutants with hyper-phosphorylated AP2 (E306K), the T160A mutants had less NCAP-1 at the nerve ring (*Figure 5*). These results are consistent with the model that NCAP-1 associates with AP2 in a phosphorylation-dependent manner.

The preferential association of NCAP-1 with the hyper-phosphorylated AP2 (E306K) compared to the phosphorylation-defective AP2 (T160A) was not simply due to differences in the extent to which these mutations generate a protease-sensitive, open AP2 complex (*Hollopeter et al., 2014*). When

the E306K and T160A mutations were both present in the same μ2 subunit, less RFP:NCAP-1 was localized to the nerve ring compared to when the E306K mutation was combined with another open AP2 mutation (R440S; *Figure 1—figure supplement 1C* and *Figure 5*) (*Hollopeter et al., 2014*). In other words, the effect of the phosphorylation site mutation was dominant, while the two open AP2 mutations were additive, with respect to NCAP-1 association. Thus, phosphorylation site mutations preclude the association of NCAP-1 with open AP2.

## Missense mutations render NECAPs functionally inactive

In the *fcho-1* suppressor screen, we isolated two independent missense mutations in the N-terminal PHear domain of NCAP-1: A29D and S84N (*Figure 1A*). To investigate the mechanism by which these mutations restore AP2 activity, we engineered them de novo into an RFP:NCAP-1 transgene in an otherwise *ncap-1* null background. These modified NECAPs failed to complement *ncap-1* mutants (*Figure 6A*), were unable to restore the closed conformation of AP2 in vivo (*Figure 6B*), and were not recruited to the nerve ring by open, hyper-phosphorylated AP2 (*Figure 6—figure supplement 1A*). Importantly, these mutant proteins were not simply unstable; by western blot analysis they appeared to be expressed at levels similar to the wild type version (*Figure 6C*). We also introduced the homologous mutations into recombinant *M. musculus* NECAP2 constructs to evaluate AP2 binding using purified components. Even though these mutant NECAPs appeared to be stable and retained the ability to bind open AP2 (E302K), they specifically failed to bind the phosphorylated AP2 core (*Figure 6—figure supplement 1B*). The in vitro pulldown assays suggest that the primary reason the NCAP-1 missense mutants are inefficiently recruited to the nerve ring by the open AP2 mutation in vivo may be that the corresponding adaptor complexes are also phosphorylated. Taken together, our results further suggest that the NCAP-1 mutants fail to inactivate AP2 because they are unable to bind the phosphorylated AP2 core.

## Discussion

In *C. elegans*, loss-of-function mutations in *ncap-1* bypass the requirement for FCHO-1 by restoring AP2 activity and promoting the accumulation of the open, phosphorylated form of AP2. These initial results were consistent with two distinct models: (1) NCAP-1 maintains the closed conformation of AP2 until FCHO-1 releases this inhibition, or (2) NCAP-1 generates the closed conformation of AP2, either directly or indirectly. As a consequence, the inactive form of AP2 predominates in the absence of allosteric activation by FCHO-1. Our subsequent biochemical and imaging data are most consistent with the latter model; NECAP homologs from a variety of organisms bind the open, phosphorylated AP2 core in vitro and restore the closed form of the complex in vivo. We also observe increased recruitment of fluorescently-tagged NCAP-1 by open, hyper-phosphorylated forms of AP2 in vivo. We propose that NECAPs act downstream of muniscin function and μ2 phosphorylation to inactivate AP2 (*Figure 7*).

### *fcho-1* suppressors disrupt the AP2 inactivation pathway

In the absence of FCHO-1, nematodes exhibit reduced fitness and AP2 dwells in a closed, hypo-phosphorylated state. Using an unbiased genetic screen, we isolated mutations that independently improve the fitness of *fcho-1* genetic nulls. We now recognize that these *fcho-1* suppressors occur in three distinct classes: dominant missense mutations that disrupt the closed conformation of AP2 (class 1) or mutate the phosphorylation site on μ2 (class 2), and recessive mutations that inactivate NCAP-1 (class 3) (*Figure 7*). These three classes of mutations appear to rescue *fcho-1* by restoring AP2 activity, but we propose that they achieve this through different mechanisms. Previously, the open AP2 mutants (class 1) appeared to behave similarly to the phosphorylation site mutants (class 2) (*Hollopeter et al., 2014*). However, this study reveals that NECAPs clearly distinguish between these two classes. Open, hyper-phosphorylated AP2 mutants (class 1) appear to recruit NECAPs in excess but must be somewhat resistant to their action in order to sustain AP2 activity in the absence of muniscins. By contrast, phosphorylation site mutants (class 2) appear to bypass the requirement for FCHO-1 by evading NECAP binding altogether. Similar to class 2, mutations that directly disrupt NCAP-1 (class 3) probably enable a basal level of AP2 activity to be sustained by complexes that succeed in attaining the open, phosphorylated state in lieu of allosteric activation by muniscins. Importantly, our data suggest that all three classes of suppressors bypass FCHO-1 by disrupting the

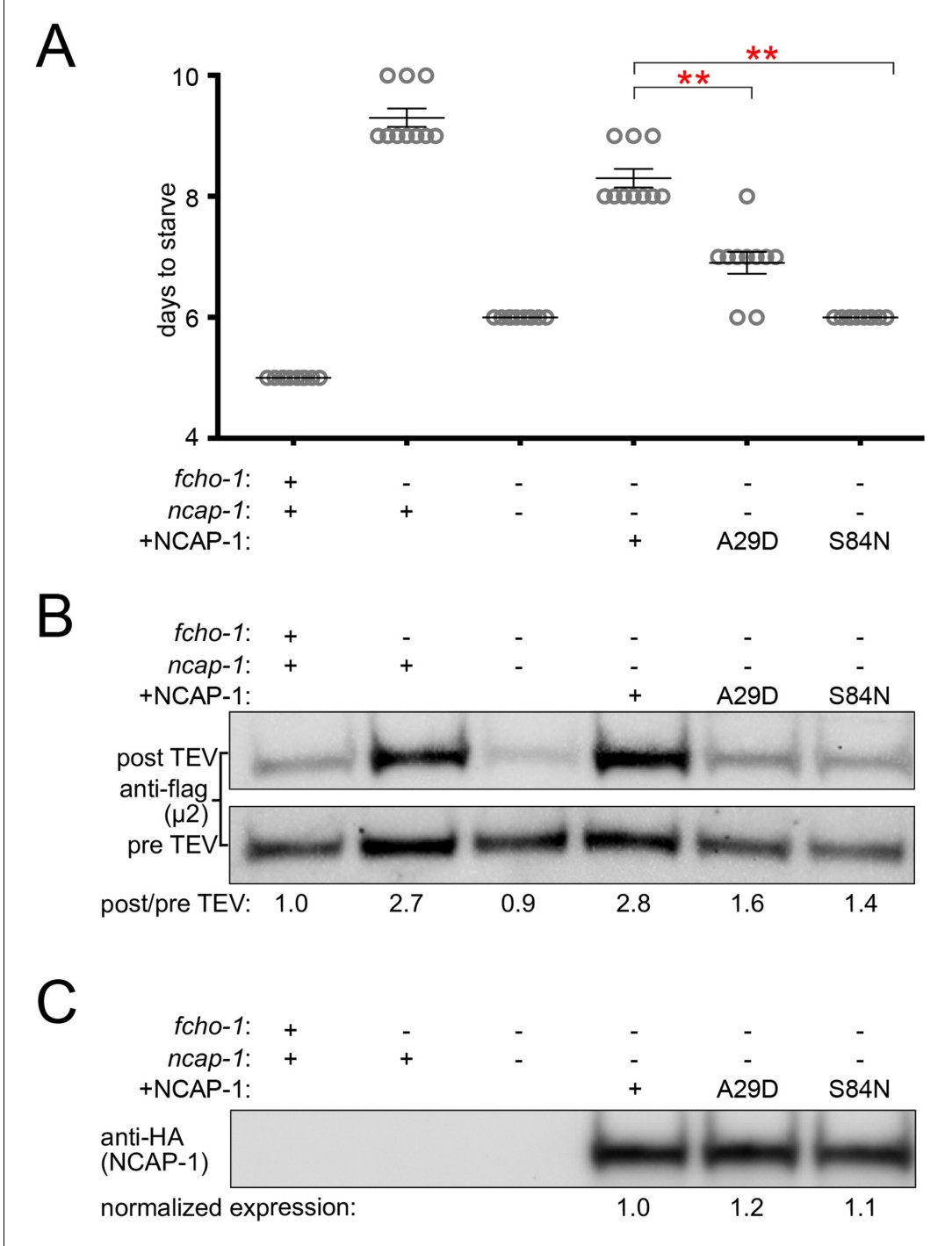

**Figure 6.** Missense mutations render NCAP-1 stable but functionally inactive. Amino acid changes isolated from the *fcho-1* suppressor screen (*Figure 1A*) were introduced into an RFP-tagged NCAP-1 transgene in *fcho-1 ncap-1* worms. +, wild type allele; -, deletion allele. (**A**) Starvation assay performed as in *Figure 1C*. Bars represent mean ±SEM for n ≥ 9 biological replicates. **p<0.001, unpaired, two-tailed T-test. (**B**) μ2 protease-sensitivity assay as in *Figure 3B*. (**C**) Western blot analysis to detect HA epitope on NCAP-1 transgenic proteins. (**B and C**) Band intensities were compared to a beta actin loading control and normalized to the *fcho-1*(+) *ncap-1*(+) ratio (**B**) or to the transgenic wild type form of NCAP-1 (**C**) (values below). Blots are representative of 2 biological replicates.

DOI: https://doi.org/10.7554/eLife.32242.008

The following figure supplement is available for figure 6:

**Figure supplement 1.** Missense mutations in NECAPs prevent association with phosphorylated forms of AP2.

*Figure 6 continued on next page*

*Figure 6 continued*

DOI: https://doi.org/10.7554/eLife.32242.009

process by which AP2 closes. The wealth of mutations in AP2 and NECAP identified in the *fcho-1* suppressor screen, combined with our biochemical and imaging data, reveal discrete steps of AP2 conformational and phosphorylation changes during its recycling (*Figure 7*).

It is curious that the unnaturally active forms of AP2 isolated from our *fcho-1* suppressor screen do not exhibit enhanced endocytosis or membrane association when the *fcho-1* gene is intact (*Hollopeter et al., 2014*) (*Figure 2*). In other words, why is there not a functional consequence associated with hyperactive AP2 in an otherwise wild type background? Perhaps there are compensatory mechanisms that prevent rampant endocytosis or, alternatively, our assays may lack the sensitivity and range necessary to detect the differences. Indeed, endocytic pits become enlarged after knockdown of NECAP1 and these structures could be a consequence of overactive AP2 (*Ritter et al., 2013*). Although vesicle size was not examined in this paper, regulating the time during which AP2 is phosphorylated and open could directly impact the size of the vesicle that gets internalized.

## AP2 phosphorylation modulates NECAP recruitment

NECAPs form stable complexes in vitro with open and phosphorylated AP2 cores, suggesting that activated AP2 may be the endogenous substrate of NECAPs. Indeed, imaging data suggest NECAP dynamics during endocytic pit formation mimic those of clathrin (*Taylor et al., 2011*), consistent with our model that AP2 activation precedes NECAP recruitment. The accumulation of phosphorylated AP2 in the absence of NCAP-1 indicates that phosphorylation may also precede NECAP recruitment but is it unclear whether the phospho-threonine is an essential feature of the NECAP-AP2 interface. Although changing the threonine to a phosphorylation-defective alanine does appear to disrupt the association of NECAPs with AP2 (*Figures 4C* and *5*), we have not excluded the possibility that this mutation simply precludes NECAP binding by altering the conformation of AP2. Indeed, we observe that NECAPs bind open AP2 cores that have not been phosphorylated (the E302K cores, *Figure 4B*) supporting the model that NECAPs bind to a face of active AP2 that does

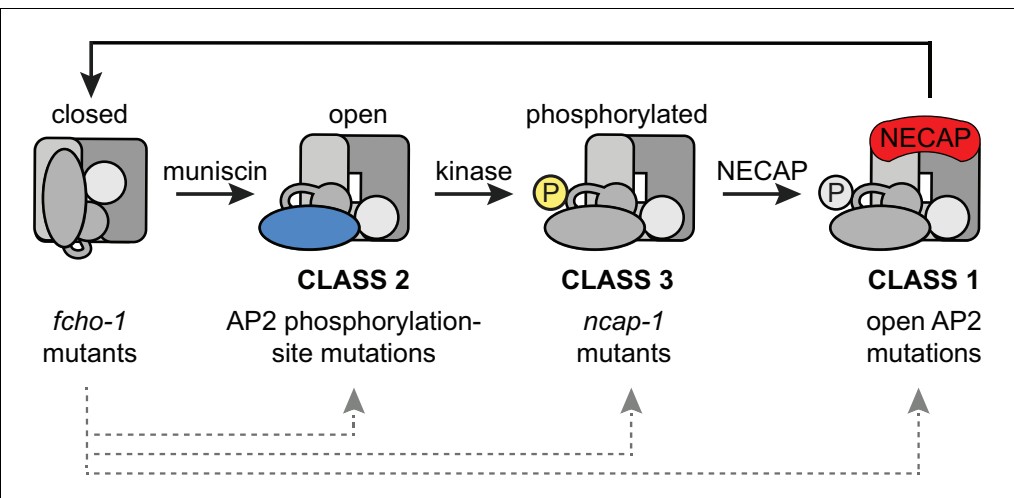

**Figure 7.** Model of AP2 activation and inactivation. Muniscins allosterically activate AP2 to form a stable association with the membrane. Open AP2 is then phosphorylated on the μ2 subunit by the AP2-associated kinase. NECAPs subsequently bind to open, phosphorylated AP2 and recycle the complex. Generation of the closed form of AP2 presumably involves dephosphorylation and disengagement from the membrane. In the absence of muniscins, AP2 activation is greatly reduced. The *fcho-1* suppressor screen isolated three classes of mutations that enable AP2 to remain active in lieu of muniscins (bottom). Each class disrupts the AP2 inactivation pathway and promotes accumulation of AP2 at discrete steps in the cycle (gray arrows).
DOI: https://doi.org/10.7554/eLife.32242.010

not include the phosphorylated threonine. However, an interaction with these constitutively open, non-phosphorylated cores in vitro may not be relevant; functionally inactive NECAPs (the A29D and S84N mutants) appear to bind these cores in vitro (*Figure 6—figure supplement 1B*), but do not associate with the equivalent, albeit hyper-phosphorylated, open AP2 mutation in vivo (*Figure 6—figure supplement 1A*). The non-phosphorylated open AP2 core may not be a physiological state of AP2 and might not accurately simulate the in vivo interaction between NECAP and AP2.

### The core function of NECAPs

While NECAPs are widely conserved across eukaryotic organisms (*Dergai et al., 2016*), the precise function of this protein family has remained enigmatic. Vertebrate NECAPs exhibit different tissue distributions and are proposed to function in mutually exclusive pathways. For example, NECAP2 was recently shown to regulate AP1 instead of AP2 (*Chamberland et al., 2016*). Despite this proposed divergence, both vertebrate NECAPs rescue loss of NCAP-1 in the nematode, as does a fungal NECAP that lacks the C-terminal α adaptin-ear binding WXXF motif. These results indicate that the capacity to negatively regulate AP2 has been conserved among NECAPs. Based on the missense mutations that disrupt NECAP activity in vivo and AP2 binding in vitro (*Figure 6* and *Figure 6—figure supplement 1*), we believe this inhibitory function may involve a direct interaction of the N-terminal PHear domain (*Ritter et al., 2007*) with the phosphorylated AP2 core.

How do NECAPs sustain the AP2 cycle? We propose the following model (*Figure 7*): muniscins such as FCHO-1 promote the open state of AP2, which is phosphorylated on the μ2 subunit by the AP2-associated kinase. NECAPs then bind the AP2 core to counteract the open state, presumably by facilitating a conformational change or dephosphorylation event later in endocytosis. This recycles AP2 back to its inactive state in the cytosol and renders the complex available for another round of endocytosis.

## Materials and methods

**Key resources table**

| Reagent type (species) or resource | Designation | Source or reference | Identifiers | Additional information |
|---|---|---|---|---|
| gene (*Caenorhabditis elegans*) | ncap-1 | NA | CELE_Y110A2AR.3 | |
| gene (*C. elegans*) | fcho-1 | NA | CELE_F56D12.6 | |
| gene (*C. elegans*) | apm-2 | NA | CEAP50, apm-2, CELE_R160.1 | |
| gene (*C. elegans*) | apa-2 | NA | apt-4, CELE_T20B5.1 | |
| strain, strain background (*C. elegans*, hermaphrodite) | N2 | NA | RRID:WB-STRAIN: N2_(ancestral)) | Wild type |
| strain, strain background (*C. elegans*, hermaphrodite) | GUN109 | this paper | | fcho-1(ox477::unc-119(+)) II ncap-1(mew31[splice donor]) II |
| strain, strain background (*C. elegans*, hermaphrodite) | GUN110 | this paper | | fcho-1(ox477::unc-119(+)) II ncap-1(mew32[A29D]) II |
| strain, strain background (*C. elegans*, hermaphrodite) | GUN111 | this paper | | fcho-1(ox477::unc-119(+)) II ncap-1(mew33[S84N]) II |
| strain, strain background (*C. elegans*, hermaphrodite) | GUN112 | this paper | | fcho-1(ox477::unc-119(+)) II ncap-1(mew34[splice donor]) II |
| strain, strain background (*C. elegans*, hermaphrodite) | GUN113 | this paper | | fcho-1(ox477::unc-119(+)) II ncap-1(mew35[splice donor]) II |
| strain, strain background (*C. elegans*, hermaphrodite) | GUN114 | this paper | | fcho-1(ox477::unc-119(+)) II ncap-1(mew36[Q107X]) II |
| strain, strain background (*C. elegans*, hermaphrodite) | GUN115 | this paper | | fcho-1(ox477::unc-119(+)) II ncap-1(mew37[splice donor]) II |
| strain, strain background (*C. elegans*, hermaphrodite) | GUN116 | this paper | | fcho-1(ox477::unc-119(+)) II ncap-1(mew38[stop lost]) II |

*Continued on next page*

*Continued*

| Reagent type (species) or resource | Designation | Source or reference | Identifiers | Additional information |
|---|---|---|---|---|
| strain, strain background (*C. elegans*, hermaphrodite) | GUN101 | this paper | | *fcho-1(ox477::unc-119(+)) II ncap-1(mew39[1.4 kb deletion]) II* |
| strain, strain background (*C. elegans*, hermaphrodite) | EG6353 | DOI: 10.7554/eLife.03648 | | *fcho-1(ox477::unc-119(+)) II; unc-119(ed3) III* |
| strain, strain background (*C. elegans*, hermaphrodite) | GUN86 | this paper | | *ncap-1(mew39[1.4 kb deletion]) II* |
| strain, strain background (*C. elegans*, hermaphrodite) | GUN59 | this paper | | *fcho-1(ox477::unc-119(+)) II mewSi2 [Pdpy-30::RFP:NCAP1 unc-119(+)] II ncap-1(mew39[1.4 kb deletion]) II* |
| strain, strain background (*C. elegans*, hermaphrodite) | EG8012 | DOI: 10.7554/eLife.03648 | | *oxSi254[Pdpy-30::APA-2::GFP unc-119(+)] II; unc-119(ed3) III* |
| strain, strain background (*C. elegans*, hermaphrodite) | EG6650 | DOI: 10.7554/eLife.03648 | | *fcho-1(ox477::unc-119(+)) II oxSi254 [Pdpy-30::APA-2::GFP unc-119(+)] II* |
| strain, strain background (*C. elegans*, hermaphrodite) | GUN98 | this paper | | *fcho-1(ox477::unc-119(+)) II oxSi254 [Pdpy-30::APA-2::GFP unc-119(+)] II ncap-1(mew39[1.4 kb deletion]) II* |
| strain, strain background (*C. elegans*, hermaphrodite) | GUN97 | this paper | | *oxSi254[Pdpy-30::APA-2::GFP unc-119(+)] II ncap-1(mew39 [1.4 kb deletion]) II* |
| strain, strain background (*C. elegans*, hermaphrodite) | EG8578 | DOI: 10.7554/eLife.03648 | | *oxSi484[Pvha-6::GFP:CD4:YASV unc-119(+)] II; apm-2(ox546[W64X]) X oxSi876[Papm-2::HA:APM-2: tev-site unc-119(+)] X* |
| strain, strain background (*C. elegans*, hermaphrodite) | EG8579 | DOI: 10.7554/eLife.03648 | | *fcho-1(ox477::unc-119(+)) II oxSi484[Pvha-6::GFP:CD4:YASV unc-119(+)] II; apm-2(ox546[W64X]) X oxSi876[Papm-2::HA:APM-2: tev-site unc-119(+)] X* |
| strain, strain background (*C. elegans*, hermaphrodite) | GUN65 | this paper | | *fcho-1(ox477::unc-119(+)) II oxSi484[Pvha-6::GFP:CD4:YASV unc-119(+)] II ncap-1(mew39 [1.4 kb deletion]) II; apm-2(ox546[W64X]) X oxSi876[Papm-2::HA:APM-2: tev-site unc-119(+)] X* |
| strain, strain background (*C. elegans*, hermaphrodite) | GUN66 | this paper | | *oxSi484[Pvha-6::GFP:CD4:YASV unc-119(+)] II ncap-1(mew39 [1.4 kb deletion]) II; apm-2(ox546[W64X]) X oxSi876 [Papm-2::HA:APM-2: tev-site unc-119(+)] X* |
| strain, strain background (*C. elegans*, hermaphrodite) | EG8557 | DOI: 10.7554/eLife.03648 | | *oxSi883[Phsp-16.41::TEV(protease) unc-119(+)] II; apm-2(ox546[W64X]) X oxSi876[Papm-2::HA:APM-2: tev-site unc-119(+)]* |
| strain, strain background (*C. elegans*, hermaphrodite) | EG8558 | DOI: 10.7554/eLife.03648 | | *fcho-1(ox477::unc-119(+)) II oxSi883[Phsp-16.41::TEV(protease) unc-119(+)] II; apm-2(ox546[W64X]) X oxSi876[Papm-2::HA:APM-2: tev-site unc-119(+)] X* |
| strain, strain background (*C. elegans*, hermaphrodite) | GUN100 | this paper | | *fcho-1(ox477::unc-119(+)) II; oxSi883[Phsp-16.41::TEVprotease unc-119(+)] II ncap-1(mew39 [1.4 kb deletion]) II; apm-2(ox546[W64X]) X oxSi876[Papm-2::HA:APM-2: tev-site unc-119(+)] X* |

*Continued*

| Reagent type (species) or resource | Designation | Source or reference | Identifiers | Additional information |
|---|---|---|---|---|
| strain, strain background (*C. elegans*, hermaphrodite) | GUN99 | this paper | | *oxSi883[Phsp-16.41::TEVprotease unc-119(+)] II ncap-1(mew39 [1.4 kb deletion]) II; apm-2(ox546[W64X]) X oxSi876[Papm-2::HA:APM-2: tev-site unc-119(+)] X* |
| strain, strain background (*C. elegans*, hermaphrodite) | EG8555 | DOI: 10.7554/eLife.03648 | | *oxSi883[Phsp-16.41::TEVprotease unc-119(+)] II; apm-2(ox546[W64X]) X oxSi877[Papm-2::3XFLAG:APM-2: tev-site unc-119(+)] X* |
| strain, strain background (*C. elegans*, hermaphrodite) | EG8556 | DOI: 10.7554/eLife.03648 | | *fcho-1(ox477::unc-119(+)) II oxSi883[Phsp-16.41::TEVprotease unc-119(+)] II; apm-2(ox546[W64X]) X oxSi877[Papm-2::3xFLAG:APM-2: tev-site unc-119(+)] X* |
| strain, strain background (*C. elegans*, hermaphrodite) | GUN96 | this paper | | *fcho-1(ox477::unc-119(+)) II oxSi883[Phsp-16.41::TEVprotease unc-119(+)] II ncap-1(mew39 [1.4 kb deletion]) II; apm-2(ox546[W64X]) X oxSi877[Papm-2::3xFLAG:APM-2: tev-site unc-119(+)] X* |
| strain, strain background (*C. elegans*, hermaphrodite) | GUN106 | this paper | | *fcho-1(ox477::unc-119(+)) II oxSi883[Phsp-16.41::TEVprotease unc-119(+)] II ncap-1(mew39 [1.4 kb deletion]) II; mewSi3 [Pdpy-30::RFP:NCAP-1 unc-119(+)] IV; apm-2(ox546[W64X]) X oxSi877 [Papm-2::3xFLAG:APM-2: tev-site unc-119(+)] X* |
| strain, strain background (*C. elegans*, hermaphrodite) | GUN91 | this paper | | *fcho-1(ox477::unc-119(+)) II oxSi883[Phsp-16.41::TEVprotease unc-119(+)] II ncap-1(mew39 [1.4 kb deletion]) II; mewSi15 [Pdpy-30::RFP:Mm_NECAP1 unc-119(+)] IV; apm-2(ox546[W64X]) X oxSi877[Papm-2::3xFLAG:APM-2: tev-site unc-119(+)] X* |
| strain, strain background (*C. elegans*, hermaphrodite) | GUN93 | this paper | | *fcho-1(ox477::unc-119(+)) II oxSi883[Phsp-16.41::TEVprotease unc-119(+)] II ncap-1(mew39 [1.4 kb deletion]) II; mewSi8 [Pdpy-30::RFP:Mm_NECAP2 unc-119(+)] IV; apm-2(ox546[W64X]) X oxSi877[Papm-2::3xFLAG:APM-2: tev-site unc-119(+)] X* |
| strain, strain background (*C. elegans*, hermaphrodite) | GUN95 | this paper | | *fcho-1(ox477::unc-119(+)) II oxSi883[Phsp-16.41::TEVprotease unc-119(+)] II ncap-1(mew39 [1.4 kb deletion]) II; mewSi17 [Pdpy-30::RFP:Ss_NECAP unc-119(+)] IV; apm-2(ox546[W64X]) X oxSi877 [Papm-2::3xFLAG:APM-2: tev-site unc-119(+)] X* |
| strain, strain background (*C. elegans*, hermaphrodite) | GUN60 | this paper | | *mewSi1[Pdpy-30::APA2:GFP unc-119(+)] I; mewSi2[Pdpy-30:: RFP:NCAP1 unc-119(+)] II ncap-1(mew39[1.4 kb deletion]) II* |
| strain, strain background (*C. elegans*, hermaphrodite) | GUN61 | this paper | | *mewSi1[Pdpy-30::APA2:GFP unc-119(+)] I; fcho-1(ox477:: unc-119(+)) II mewSi2[Pdpy-30:: RFP:NCAP1 unc-119(+)] II ncap-1(mew39[1.4 kb deletion]) II* |

*Continued on next page*

*Continued*

| Reagent type (species) or resource | Designation | Source or reference | Identifiers | Additional information |
|---|---|---|---|---|
| strain, strain background (*C. elegans*, hermaphrodite) | GUN62 | this paper | | *mewSi1[Pdpy-30::APA2:GFP unc-119(+)] I; fcho-1(ox477::unc-119(+)) II mewSi2[Pdpy-30::RFP:NCAP1 unc-119(+)] II ncap-1(mew39 [1.4 kb deletion]) II; apm-2(ox562[E306K]) X* |
| strain, strain background (*C. elegans*, hermaphrodite) | GUN53 | this paper | | *mewSi1[Pdpy-30::APA2:GFP unc-119(+)] I; fcho-1(ox477:: unc-119(+)) II mewSi2[Pdpy-30:: RFP:NCAP1 unc-119(+)] II ncap-1(mew39[1.4 kb deletion]) II; apm-2(mew44[T160A]) X* |
| strain, strain background (*C. elegans*, hermaphrodite) | GUN55 | this paper | | *mewSi1[Pdpy-30::APA2:GFP unc-119(+)] I; fcho-1(ox477:: unc-119(+)) II mewSi2[Pdpy-30:: RFP:NCAP1 unc-119(+)] II ncap-1(mew39[1.4 kb deletion]) II; apm-2(ox562[E306K]+mew46[T160A]*) X* |
| strain, strain background (*C. elegans*, hermaphrodite) | GUN56 | this paper | | *mewSi1[Pdpy-30::APA2:GFP unc-119(+)] I; fcho-1(ox477:: unc-119(+)) II mewSi2[Pdpy-30:: RFP:NCAP1 unc-119(+)] II ncap-1(mew39[1.4 kb deletion]) II; apm-2(ox562[E306K]+mew47[R440S]*) X* |
| strain, strain background (*C. elegans*, hermaphrodite) | GUN128 | this paper | | *fcho-1(ox477::cb-unc-119(+)) II oxSi883[Phsp-16.41::TEVprotease Cb_unc-119(+)] II ncap-1(mew39 [1.4 kb deletion]) II; mewSi25[RFP:NCAP-1(A29D)*] IV; apm-2(ox546[W64X]) X oxSi877[Papm-2::3xFLAG:APM-2: tev-site Cb_unc-119(+)] X* |
| strain, strain background (*C. elegans*, hermaphrodite) | GUN135 | this paper | | *fcho-1(ox477::cb-unc-119(+)) II oxSi883[Phsp-16.41::TEVprotease Cb_unc-119(+)] II ncap-1(mew39 [1.4 kb deletion]) II; mewSi35[RFP:NCAP-1(S84N)*] IV; apm-2(ox546[W64X]) X oxSi877 [Papm-2::3xFLAG:APM-2: tev-site Cb_unc-119(+)] X* |
| strain, strain background (*C. elegans*, hermaphrodite) | GUN127 | this paper | | *mewSi1[APA2:GFP] I; fcho-1(ox477:: unc-119(+)) II mewSi24[RFP: NCAP1(A29D)*] II ncap-1(mew39[1.4 kb deletion]) II; apm-2(ox562[E306K])X* |
| strain, strain background (*C. elegans*, hermaphrodite) | GUN122 | this paper | | *mewSi1[APA2:GFP] I; fcho-1(ox477:: unc-119(+)) II mewSi31[RFP: NCAP1(S84N)*] II ncap-1(mew39[1.4 kb deletion]) II; apm-2(ox562[E306K])X* |
| genetic reagent (*C. elegans*) | *fcho-1(ox477::unc-119(+))* | DOI: 10.7554/eLife.03648 | | |
| genetic reagent (*C. elegans*) | *ncap-1(mew31[splice donor])* | this paper | | *fcho-1* suppressor |
| genetic reagent (*C. elegans*) | *ncap-1(mew32[A29D])* | this paper | | *fcho-1* suppressor |
| genetic reagent (*C. elegans*) | *ncap-1(mew33[S84N])* | this paper | | *fcho-1* suppressor |
| genetic reagent (*C. elegans*) | *ncap-1(mew34[splice donor])* | this paper | | *fcho-1* suppressor |
| genetic reagent (*C. elegans*) | *ncap-1(mew35[splice donor])* | this paper | | *fcho-1* suppressor |
| genetic reagent (*C. elegans*) | *ncap-1(mew36[Q107X])* | this paper | | *fcho-1* suppressor |
| genetic reagent (*C. elegans*) | *ncap-1(mew37[splice donor])* | this paper | | *fcho-1* suppressor |
| genetic reagent (*C. elegans*) | *ncap-1(mew38[stop lost])* | this paper | | *fcho-1* suppressor |

*Continued on next page*

*Continued*

| Reagent type (species) or resource | Designation | Source or reference | Identifiers | Additional information |
|---|---|---|---|---|
| genetic reagent (*C. elegans*) | ncap-1(mew39[1.4 kb deletion]) | this paper | | *fcho-1* suppressor |
| genetic reagent (*C. elegans*) | mewSi2[Pdpy-30::RFP:NCAP1 unc-119(+)] | this paper | | Generated with MosSCI |
| genetic reagent (*C. elegans*) | mewSi3[Pdpy-30::RFP:NCAP-1 unc-119(+)] | this paper | | Generated with MosSCI |
| genetic reagent (*C. elegans*) | mewSi15[Pdpy-30::RFP: Mm_NECAP1 unc-119(+)] | this paper | | Generated with MosSCI |
| genetic reagent (*C. elegans*) | mewSi8[Pdpy-30::RFP: Mm_NECAP2 unc-119(+)] | this paper | | Generated with MosSCI |
| genetic reagent (*C. elegans*) | mewSi17[Pdpy-30::RFP: Ss_NECAP unc-119(+)] | this paper | | Generated with MosSCI |
| genetic reagent (*C. elegans*) | oxSi254[Pdpy-30::APA-2:: GFP unc-119(+)] | DOI: 10.7554/eLife.03648 | | |
| genetic reagent (*C. elegans*) | mewSi1[Pdpy-30:: APA-2::GFP unc-119(+)] | this paper | | Generated with MosSCI |
| genetic reagent (*C. elegans*) | oxSi484[Pvha-6::GFP:CD4: YASV unc-119(+)] | DOI: 10.7554/eLife.03648 | | |
| genetic reagent (*C. elegans*) | oxSi883[Phsp-16.41:: TEV(protease) unc-119(+)] | DOI: 10.7554/eLife.03648 | | |
| genetic reagent (*C. elegans*) | oxSi876[Papm-2::HA:APM-2: tev-site unc-119(+)] | DOI: 10.7554/eLife.03648 | | |
| genetic reagent (*C. elegans*) | oxSi880[Papm-2::HA:APM-2(E306K): tev-site unc-119(+)] | DOI: 10.7554/eLife.03648 | | |
| genetic reagent (*C. elegans*) | oxSi878[Papm-2::HA:APM-2(T160A): tev-site unc-119(+)] | DOI: 10.7554/eLife.03648 | | |
| genetic reagent (*C. elegans*) | oxSi877[Papm-2::3xFLAG: APM-2:tev-site unc-119(+)] | DOI: 10.7554/eLife.03648 | | |
| genetic reagent (*C. elegans*) | apm-2(ox562[E306K] +mew46[T160A]) | this paper | | *mew46*[T160A] generated by CRISPR |
| genetic reagent (*C. elegans*) | apm-2(ox562[E306K] +mew47[R440S]) | this paper | | *mew47*[R440S] generated by CRISPR |
| genetic reagent (*C. elegans*) | mewSi25[RFP:NCAP-1(A29D] | this paper | | Generated by CRISPR |
| genetic reagent (*C. elegans*) | mewSi35[RFP:NCAP-1(S84N)] | this paper | | Generated by CRISPR |
| genetic reagent (*C. elegans*) | mewSi24[RFP:NCAP1(A29D)] | this paper | | Generated by CRISPR |
| genetic reagent (*C. elegans*) | mewSi31[RFP:NCAP1(S84N)] | this paper | | Generated by CRISPR |
| cell line (*Homo sapiens*, female) | HEK293 | ATCC | RRID:CVCL_0045 | |
| transfected construct (*H. sapiens*) | pGH500, in HEK239 cells | this paper | | Cloning described in 'Tissue culture pulldowns' |
| transfected construct (*H. sapiens*) | pGH501, in HEK239 cells | this paper | | Cloning described in 'Tissue culture pulldowns' |
| transfected construct (*H. sapiens*) | pGH502, in HEK239 cells | this paper | | Cloning described in 'Tissue culture pulldowns' |
| antibody | mouse monoclonal anti-adaptin α | BD Biosciences | Cat# 610501, RRID:AB_397867 | (1:500) |
| antibody | rabbit polyclonal anti-AP2B1 | Abcam | 151961, RRID: AB_2721072 | (1:1000) |
| antibody | rabbit monoclonal anti-AP2M1 phospho T156 | Abcam | Cat# 109397, RRID:AB_10866362 | (1:1000) |
| antibody | rabbit monoclonal anti-AP2S1 | Abcam | Cat# 128950, RRID:AB_11140842 | (1:4000) |

*Continued on next page*

*Continued*

| Reagent type (species) or resource | Designation | Source or reference | Identifiers | Additional information |
|---|---|---|---|---|
| antibody | mouse monoclonal anti-flag | Sigma-Aldrich | Cat# F3165, RRID:AB_259529 | (1:1000) |
| antibody | mouse monoclonal anti-tubulin | Sigma-Aldrich | Cat# T5168, RRID:AB_477579 | (1:2000) |
| antibody | rabbit polyclonal anti-histone H3 | Abcam | Cat# 1791, RRID:AB_302613 | (1:4000) |
| antibody | rabbit polyclonal anti-beta actin | Abcam | Cat# 8227, RRID:AB_2305186 | (1:1000) |
| antibody | goat anti-mouse Alexa Fluor 488 | Life Technologies | Cat # A11029, RRID:AB_2534088 | (1:4000) |
| antibody | goat anti-rabbit Alexa Fluor 647 | Life Technologies | Cat# A21244, RRID:AB_10562581 | (1:2000) |
| antibody | goat anti-rabbit StarBright Blue 700 | BioRad | Cat# 12004161, RRID: AB_2721073 | (1:5000) |
| antibody | goat anti-mouse IRDye 800CW | LI-COR | Cat# 925–32210, RRID:AB_2687825 | (1:20000) |
| antibody | rat monoclonal anti-HA-Horseradish peroxidase (HRP) | Roche | Cat# 12013819001 RRID:AB_390917 | (1:500) |
| recombinant DNA reagent | pEP29 | this paper | | Cloning described in 'C. elegans NECAP transgenes' |
| recombinant DNA reagent | pEP41 | this paper | | Cloning described in 'C. elegans NECAP transgenes' |
| recombinant DNA reagent | pEP58 | this paper | | Cloning described in 'C. elegans NECAP transgenes' |
| recombinant DNA reagent | pEP71 | this paper | | Cloning described in 'C. elegans NECAP transgenes' |
| recombinant DNA reagent | pGH495 | this paper | | Cloning described in 'C. elegans NECAP transgenes' |
| recombinant DNA reagent | pGH505 | this paper | | Cloning described in 'C. elegans NECAP transgenes' |
| recombinant DNA reagent | pGB19 | this paper | | Cloning described in 'Recombinant AP2 cores' |
| recombinant DNA reagent | pGB21 | this paper | | Cloning described in 'Recombinant AP2 cores' |
| recombinant DNA reagent | pGB27 | this paper | | Cloning described in 'Recombinant NECAPs' |
| recombinant DNA reagent | pGB28 | this paper | | Cloning described in 'Recombinant NECAPs' |
| recombinant DNA reagent | pGB29 | this paper | | Cloning described in 'Recombinant NECAPs' |
| recombinant DNA reagent | pGB31 | this paper | | Cloning described in 'Recombinant AP2 cores' |
| recombinant DNA reagent | pGB81 | this paper | | Cloning described in 'Recombinant NECAPs' |
| recombinant DNA reagent | pEP82 | this paper | | Cloning described in 'Recombinant AP2 cores' |
| recombinant DNA reagent | pGB91 | this paper | | Cloning described in 'Recombinant NECAPs' |
| recombinant DNA reagent | pGB94 | this paper | | Cloning described in 'Recombinant NECAPs' |
| sequence-based reagent | pGH494 | DOI: 10.7554/eLife.03648 | | |

*Continued on next page*

*Continued*

| Reagent type (species) or resource | Designation | Source or reference | Identifiers | Additional information |
|---|---|---|---|---|
| sequence-based reagent | pGH503 | this paper | | Cloning described in 'Recombinant NECAPs' |
| sequence-based reagent | pGH504 | this paper | | Cloning described in 'Recombinant AP2 cores' |
| sequence-based reagent | oGH678 | this paper | | CGATAGAGAAGGCTTCAACACAC |
| sequence-based reagent | oGH679 | this paper | | AGGTATTCAGACATTTTTC AAATGAAAATCTAC |
| sequence-based reagent | oGH680 | this paper | | CAGTCAAAAAATGC GATAAAAGTACGG |
| sequence-based reagent | oGH681 | this paper | | GGACAGGAAATTTC AATAAATTAGCGATG |
| sequence-based reagent | oEP366 | this paper | | AACGGGCGGTAGT GGAGGCACTGGTATG GGAGATTACGAGAACGTTTTAATG |
| sequence-based reagent | oEP367 | this paper | | TATCACCACTTTGTACAAGAAAGCT GGGTCTAGAAATCTAATAAA TTGCCAGACGTCG |
| sequence-based reagent | oEP407 | this paper | | GAGGAACGGGCGGTAGTGGAG GCACTGGTATGGAGGAGAGTG AGTACGAGTCTGTTCTGT |
| sequence-based reagent | oEP408 | this paper | | TCACCACTTTGTACAAGAAAGC TGGGTCTAGAACTGG ACCCAGCCGGTG |
| sequence-based reagent | oEP409 | this paper | | GGAGGAACGGGCGGTAGTGG AGGCACTGGTATGGCGGCA GAGCTGGAATATG |
| sequence-based reagent | oEP410 | this paper | | TCACCACTTTGTACAAGAAA GCTGGGTCTAAAACTGGA CCCAGTTAGATGGCTGTG |
| sequence-based reagent | oEP391 | this paper | | ACGTCGTGACTGGGAAAACCC |
| sequence-based reagent | oEP392 | this paper | | GCCAGGGTTTTCCCAGTCA CGACGTTGATCATTGGCA TGCTGAAATATTC |
| sequence-based reagent | oGH526 | this paper | | ATGGTTGTGTCGAAAGGCGA |
| sequence-based reagent | oGH528 | this paper | | ACCAGTGCCTCCACTACCG CCCGTTCCTCCTGTGCCACC TTTGTACAGTTCATCCATTCC |
| sequence-based reagent | oGH698 | this paper | | GGGGACAAGTTTGTACAAAA AAGCAGGCTCAAAAATGG TTGTGTCGAAAGGCGA |
| sequence-based reagent | oGH731 | this paper | | GTGACATTAAAGTC AAAAGCATCTCCTC |
| sequence-based reagent | oGH733 | this paper | | GGGGACAAGTTTGTACAAAA AAGCAGGCTCAAAAATGGGA GATTACGAGAACGTTTTAAT |
| sequence-based reagent | oGH734 | this paper | | GGGGACCACTTTGTACAAGAA AGCTGGGTTTAGAAATCTAAT AAATTGCCAGACGTC |
| sequence-based reagent | oGH736 | this paper | | GAGGAGATGCTTTTGAC TTTAATGTCAC |
| sequence-based reagent | oGH738 | this paper | | GCGGTAGTGGAGGCACTG GTATGGGAGATTACGAG AACGTTTTAATG |
| sequence-based reagent | oGH1011 | this paper | | TAGACCCAGCTTTCTTGTA CAAAGTGGTGATA |

*Continued on next page*

*Continued*

| Reagent type (species) or resource | Designation | Source or reference | Identifiers | Additional information |
|---|---|---|---|---|
| sequence-based reagent | oGH1012 | this paper | | ACCAGTGCCTCCAC TACCGCCCGTT |
| sequence-based reagent | oGH953 | this paper | | CATGCTTCCGCCGGTACCT |
| sequence-based reagent | oGH954 | this paper | | GTTTAAACCCGCTGATCAGCCT |
| sequence-based reagent | oGH955 | this paper | | GTGGAGGTACCGGCGGAAG CATGGGAGATTACGAG AACGTTTTAATG |
| sequence-based reagent | oGH956 | this paper | | GCTGATCAGCGGGTTTAAACTT AGAAATCTAATAAATT GCCAGACGTCG |
| sequence-based reagent | oGH957 | this paper | | GTGGAGGTACCGGCGGAA GCATGGCGGCAGAGCTGGAA |
| sequence-based reagent | oGH958 | this paper | | GATCAGCGGGTTTAAACTT AAAACTGGACCCAGTTAGATGGC |
| sequence-based reagent | oGH959 | this paper | | GTGGAGGTACCGGCGGAA GCATGGAGGAGAGTGAGTACGAGT |
| sequence-based reagent | oGH960 | this paper | | GCTGATCAGCGGGTTTAAA CTTAGAACTGGACCCAGCCGG |
| sequence-based reagent | oEP13 | this paper | | TAATTAACCTAGGCTGCTGCCACC |
| sequence-based reagent | oEP17 | this paper | | AAGAAGGAGATATACATAT GAAGAAGTTTTTCGACTCCAG |
| sequence-based reagent | oEP18 | this paper | | GGCAGCAGCCTAGGTTAATT ACTGTACATTTGGAACGGGGC |
| sequence-based reagent | oGB24 | this paper | | CGCCGCCAGCCAATCTGCCCA GCCACCTGGCTGGTGA TCTGGGACTGTTC |
| sequence-based reagent | oGB26 | this paper | | ATGAATAAGCLCTCCGATCA TCATATGTATATCTCCTTCTTATA |
| sequence-based reagent | oGB27 | this paper | | GCATTTATGAAACCCGCT GCTAATTAACCTAGGC TGCTGCCACCG |
| sequence-based reagent | oGB28 | this paper | | ATGATCGGAGGCTTATTCATCT |
| sequence-based reagent | oGB29 | this paper | | GCAGCGGGTTTCATAAATGCCA |
| sequence-based reagent | oGB33 | this paper | | GGGCAGATTGGCTGGCG GCGAGAAGGCATCAAGTA |
| sequence-based reagent | oGB34 | this paper | | AGCAAGAGTCTGGTGCCGCG CGGCAGCGGTAAGCAGTC GATCGCCATTGATG |
| sequence-based reagent | oGB35 | this paper | | CTGCTTACCGCTGCCGCG CGGCACCAGACTCTTGCT TGTTTCATCAGCTGTG |
| sequence-based reagent | oGB47 | this paper | | TGAGATCCGGCTGC TAACAAAGCC |
| sequence-based reagent | oGB48 | this paper | | TTTAAGAAGGAGATATACA TATGGCAGAAATCGGTACTGG |
| sequence-based reagent | oGB49 | this paper | | AGTGCATCTCCCGTGATGC AGAAATCTAATAAATTGCCA |
| sequence-based reagent | oGB50 | this paper | | AGTGCATCTCCCGTGATGC AAAACTGGACCCAGTTAGATGGC |
| sequence-based reagent | oGB51 | this paper | | AGTGCATCTCCCGTGATGCA GAACTGGACCCAGCCGGTGC |
| sequence-based reagent | oGB52 | this paper | | TGCATCACGGGAGATGCACT |

*Continued on next page*

*Continued*

| Reagent type (species) or resource | Designation | Source or reference | Identifiers | Additional information |
|---|---|---|---|---|
| sequence-based reagent | oGB53 | this paper | | GTTAGCAGCCGGATCTCAGT GGTGATGATGGTGATGTTG AAGCTGCCACAAGGCAGG |
| sequence-based reagent | oGB174 | this paper | | AGTGCATCTCCCGTGATGC AGCTTCCGCCGGTACCTCCAC |
| sequence-based reagent | oGH338 | DOI: 10.7554/eLife.03648 | | CATATGTATATCTCCTTCTT ATACTTAACTAATATAC TAAGATG |
| sequence-based reagent | oEP642 | this paper | | CCGATATCCACGGT TGGTGGCCCG |
| sequence-based reagent | oEP643 | this paper | | ACCAACCGTGGATATCGGG ACTCAGAATGGCAACT GGACCAGC |
| sequence-based reagent | oEP644 | this paper | | TCTAGATACTTCGTCATC CGAATTGAAGATGGA |
| sequence-based reagent | oEP645 | this paper | | AATTCGGATGACGAAGT ATCTAGAGTTGTCTGT CACACTCTCCACTGCG |
| sequence-based reagent | oGH847 | this paper | | CCAAACTGAAGGTCAAGGTGGTC |
| sequence-based reagent | oGH848 | this paper | | CCTTGACCTTCAGTTTGGTGCGC |
| sequence-based reagent | oGH853 | DOI: 10.7554/eLife.03648 | | TAATTAACCTAGGC TGCTGCCACCG |
| sequence-based reagent | oGH1204 | this paper | | GTTAATTAAAACAG ATGCACGACGGTT |
| sequence-based reagent | oGH1205 | this paper | | GTGCATCTGTTTTAATTAACA TGGAGGAGAGTGAGTACGAGT |
| sequence-based reagent | oGH1206 | this paper | | GCAGCAGCCTAGGTTAATTA GAACTGGACCCAGCCGG |
| sequence-based reagent | oGH1227 | this paper | | GGAAGTTCTGTTCCAGG GGCCCGGGTCCGG CATGTCCCCT |
| sequence-based reagent | oGH1228 | this paper | | GCCCCTGGAACAGAACTTC CAGGCCGGATCCGCCCTTCTT |
| sequence-based reagent | oGH1231 | this paper | | CATATGTATATCTCCTT CTTAAAGTTAAAC |
| sequence-based reagent | oGH1246 | this paper | | CCGCTGAGCAATAACT AGCATAAC |
| sequence-based reagent | oGH1247 | this paper | | CTAATGCAGGAGTCGCATAAGG |
| sequence-based reagent | oGH1249 | this paper | | GTTATGCTAGTTAT TGCTCAGCGG |
| sequence-based reagent | oGH1250 | this paper | | TTATGCGACTCCTGCATTAG GCGCGAGGCAGGATCTCG |
| sequence-based reagent | rEP360 | this paper | | Gene-specific target of crRNA: TGAAGTGTCTCGTAACAAGA |
| sequence-based reagent | rGB156 | this paper | | Gene-specific target of crRNA: CAAATCACGTCTCAAGTGAC |
| sequence-based reagent | rGB155 | this paper | | Gene-specific target of crRNA: TTGGGTGAAGTTCTAGCATC |
| sequence-based reagent | rEP254 | this paper | | Gene-specific target of crRNA: CGGGCTGTCGAGGTTCCAGT |
| sequence-based reagent | rEP676 | this paper | | Gene-specific target of crRNA: CACAAAATATCGAGAACTAT |

*Continued on next page*

*Continued*

| Reagent type (species) or resource | Designation | Source or reference | Identifiers | Additional information |
|---|---|---|---|---|
| sequence-based reagent | rEP700 | this paper | | Gene-specific target of crRNA: CCCTGGCAACGCAATTGAGG |
| sequence-based reagent | oGB154 | this paper | | TCCCATTGGTTCGcGAAGTGTCT CGTAACAAGATGaAAGTTAA GGTATTTCACTTGTCAC |
| sequence-based reagent | oGB159 | this paper | | TTCGTTACATTGGAcGATCG GGACTGTATGAAACtAGcTGC TAGAACTTCACCCAACCCT |
| sequence-based reagent | oGB130 | this paper | | GGAGCAGTCACAAATCACGT CTCAAGTtGCCGGCCAAATT GGATGGCGTCGGGAGGGTAT |
| sequence-based reagent | oEP674 | this paper | | TTCGCTATAAAATCCCTATTT TTCAGAGatGCgGACTGGAAC CTCGACAGCCCGGCTTGG |
| sequence-based reagent | oEP680 | this paper | | CCGCCGATCGGAACCAGCGG TCATAAAGatGCgGACTGGA ACCTCGACAGCCCGGCTTGG |
| sequence-based reagent | oEP701 | this paper | | GCCCGATCGATGCGCACCC TGGCAACGCAATTGAGGCcG TTTCgGATaacTCTaGATATTT TGTGATTCGTTTGCAG |
| peptide, recombinant protein | AcTEV Protease | Invitrogen | 12575015 | |
| software, algorithm | GraphPad Prism (version 7.0 c for Mac) | GraphPad Software, www.graphpad.com | RRID:SCR_002798 | |
| software, algorithm | Fiji | doi:10.1038/ nmeth.2019 | RRID:SCR_002285 | |

*allele generated by CRISPR

## Strains

*C. elegans* were maintained using standard procedures (*Brenner, 1974*) on nematode growth medium (NGM) plates seeded with *E. coli* (OP50). For a complete list of strains, see *Supplementary file 1A*.

## Data analysis

Unpaired, parametric, two-tailed T-tests were performed using GraphPad Prism (version 7.0 c for Mac, GraphPad Software, La Jolla, CA, USA, www.graphpad.com).

## Identification of *ncap-1* mutants

Our initial mutagenesis of *fcho-1* null animals (*Hollopeter et al., 2014*) yielded four recessive suppressors in the same complementation group. Single nucleotide polymorphism mapping (*Davis et al., 2005*) placed them on chromosome II and whole genome sequencing revealed four independent mutations in the *ncap-1* gene (*mew31*, *mew36*, *mew38*, and *mew39*). Additional alleles were identified among subsequent *fcho-1* suppressors by amplifying and sequencing the coding segments of *ncap-1* with primer pairs oGH678-9 and oGH680-1. Strains and oligonucleotides are listed in *Supplementary file 1*.

## Starvation assay

The starvation assay was performed as previously described (*Hollopeter et al., 2014*) except that the assay was performed at 25℃ for *Figure 1C*.

## Preparation of worms for microscopy

For confocal fluorescence microscopy, worms were mounted on 8–10% agarose pads in 3 µL of a 1:1 mix of a 1 µm polystyrene bead slurry (Polysciences, Warrington, PA) and 2X PBS pH 7.4 (*Kim et al.,*

*2013*). For differential interference contrast microscopy (*Figure 1B*), worms were mounted on 5% agarose pads in PBS pH 7.4 with 20 mM sodium azide.

## AP2 localization and FRAP analysis in coelomocytes

Worms expressing APA-2:GFP (*oxSi254*) were imaged as previously described (*Hollopeter et al., 2014*). Strains are listed in *Supplementary file 1A*.

## Cargo assay

The cargo assay was performed essentially as previously described (*Hollopeter et al., 2014*). Worms expressing an artificial cargo (*oxSi484*) were imaged on a Ziess LSM 880 confocal microscope (Biotechnology Resource Center, Cornell University, Ithaca, NY) with a 40x water immersion objective. All strains were imaged in one session with the same laser settings. Images were analyzed in Fiji (*Schindelin et al., 2012*). A user defined line was drawn along the membrane between intestinal segments 2 and 3, or segments 3 and 4, and the average pixel intensity was measured along the line. Strains are listed in *Supplementary file 1A*.

## TEV protease-sensitivity assay

The TEV assay was performed essentially as previously described (*Hollopeter et al., 2014*). Post TEV samples were collected at 5–6 hr following heatshock (34°C, 1 hr). Each sample represents 100 L4 hermaphrodites lysed in 1X Bolt LDS Sample Buffer (Invitrogen, Carlsbad, CA) containing fresh dithiothreitol (DTT; ~100 mM) by sonication (1 s pulses at 90–95% amplitude for 2–3 min) in a cup horn (Branson Ultrasonics Corporation, Danbury, CT) that was chilled to 4°C. Samples were heated for 10 min at 70°C prior to gel electrophoresis. All samples were re-sonicated following the 70°C denaturation step if any exhibited excessive viscosity. Strains are listed in *Supplementary file 1A*.

## Western blots and SYPRO staining

Precast polyacrylamide gels (Bolt 4–12% Bis-Tris, Invtrogen) were used for all SDS PAGE experiments. For western blot analysis, proteins were transferred from gels to PVDF Immobilon membranes (Merck Millipore, Tullagreen, Carrigtwohill, Co. Cork, Ireland) using the Pierce Power Blot Cassette system (Thermo Scientific, Rockford, IL). All blocking and antibody incubations (except anti-HA, see below) occurred in Odyssey Blocking Buffer (LI-COR, Lincoln, NE). Primary antibodies and dilutions included mouse anti-adaptin α (1:500, BD Biosciences, San Jose, CA, 610501), rabbit anti-AP2B1 (1:1000, Abcam, Cambridge, MA, 151961), rabbit anti-AP2M1 phospho T156 (1:1000, Abcam 109397), rabbit anti-AP2S1 (1:4000, Abcam 128950), mouse anti-flag (1:1000, Sigma-Aldrich F3165), mouse anti-tubulin (1:2000, Sigma-Aldrich T5168), rabbit anti-histone H3 (1:4000, Abcam 1791), and rabbit anti-beta actin (1:1000, Abcam, 8227). Secondary antibodies included goat anti-mouse Alexa Fluor 488 (1:4000, Life Technologies, Eugene, OR, A11029), goat anti-rabbit Alexa Fluor 647 (1:2000, Life Technologies, A21244), StarBright Blue 700 goat anti-rabbit (1:5000, BioRad, Hercules, CA, 12004161), and goat anti-mouse IRDye 800CW (1:20000, LI-COR, 925–32210).

Blots probed with the rat anti-HA-Horseradish peroxidase (HRP) antibody (1:500, Roche, Mannheim, Germany, 12013819001) were blocked in Tris Buffered Saline + 0.01% Tween 20 (TBST) with 5% nonfat dry milk. The antibody was diluted in TBST with 1% milk and incubation occurred at room temperature for 1 hr. SuperSignal West Dura Extended Duration Substrate (Thermo Scientific) was used to detect peroxidase.

Primary antibody incubations occurred for 1 hr at room temperature or overnight at 4°C. Secondary antibody incubations occurred at room temperature for 30 min – 1 hr in the dark. TBST was used for all washes. Blots were imaged using Bio-Rad ChemiDoc MP systems and band intensities were quantified using the associated ImageLab software.

We used SYPRO Ruby Protein Gel stain (Lonza, Rockland, ME) to visualize the in vitro pulldowns in *Figure 4B* and in *Figure 6—figure supplement 1B*. Gels were fixed in 50% methanol/7% acetic acid for 30 min and stained with SYPRO for 4–16 hr at room temperature in the dark with gentle agitation. Gels were then washed 2 × 15 min in 10% methanol/7% acetic acid and then 2 × 5 min in water before imaging on a Bio-Rad ChemiDoc MP system.

## *C. elegans* NECAP transgenes

We generated *C. elegans* targeting vectors for expression of NECAPs as single-copy transgenes (*Frøkjaer-Jensen et al., 2008*). For *Figure 1B and C*, *Figure 5*, and *Figure 6—figure supplement 1A*, a mini-gene encoding NCAP-1 was constructed using the Multisite Gateway System (Invitrogen). The first 4 exons of the *ncap-1* gene were amplified with primer pair oGH731 + 3, while the other half of the coding sequence was amplified from cDNA using primer pair oGH734 + 6. The two amplicons were recombined using the Gibson assembly reaction (*Gibson et al., 2009*) and then recombined with the [1-2] donor vector using BP clonase (Invitrogen). The entry clone was amplified with oGH698 + 738 and a worm codon-optimized TagRFP-T (Ed Boyden, MIT, Cambridge, MA) was amplified with oGH526 + 8 and inserted upstream of the *ncap-1* coding sequences using Gibson assembly. The resulting [1-2] entry clone (pGH505) was recombined with a [4-1] entry containing the ubiquitous dpy-30 promoter, the unc-54 3'UTR in a [2-3] entry and the [4-3] destination vector pCFJ150 (Christian Frøkjær-Jensen, University of Utah) using LR clonase (Invitrogen) to generate the MosSCI targeting vector (pGH495) that was injected into worm strain EG6699 (*Frøkjær-Jensen et al., 2012*).

For *Figure 3*, the coding sequences of NECAPs were amplified from plasmid templates using the following primers: oEP366-7 for *C. elegans* NCAP-1 (NM_061997.5), oEP409-10 for *M. musculus* NECAP1 (BK000656.1) and oEP407-8 for *M. musculus* NECAP2 (BK000657.1). The coding sequence of NECAP from *Sphaerobolus stellatus* (Cannonball Fungus protein KIJ44287) was synthesized as a gBlock (IDT, Coralville, IA). These DNA fragments were then assembled in a MosSCI targeting vector backbone generated as two amplicons from pGH486 (*Hollopeter et al., 2014*), using primer pairs oGH1011 + oEP392 and oGH1012 + oEP391, in a three-piece Gibson assembly reaction. The plasmids generated were: pEP29 for *C. elegans* NCAP-1, pEP58 for *M. musculus* NECAP1, pEP41 for *M. musculus* NECAP2, and pEP71 for *S. stellatus* NECAP. These included targeting sequences corresponding to the *cxTi10816* locus and were inserted into the genomes of EG6703 worms as described (*Frøkjær-Jensen et al., 2012*). The resulting transgenes drive expression of RFP-tagged NECAPs from a ubiquitous *C. elegans* promoter (*Pdpy-30*). Strains, plasmids, and oligonucleotides are listed in *Supplementary file 1*.

## Tissue culture pulldowns

The heterologous expression of affinity-tagged proteins in HEK293 cells followed by HaloTag isolation from cell lysates and western blots analysis of the purified proteins was performed as described (*Hollopeter et al., 2014*). Mammalian vectors for the expression of HaloTag fusions with the worm and mouse NECAPs were generated by amplifying the coding sequences from cDNA and inserting them by Gibson assembly into a custom-built HaloTag expression vector (*Banks et al., 2014*) that was amplified using oGH953-4. The *C. elegans* NCAP-1 (NM_061997.5; amplicon oGH955-6) expression vector is pGH500, while the *M. musculus* NECAP1 (BK000656.1; amplicon oGH957-8) is pGH501, and *M. musculus* NECAP2 (BK000657.1; amplicon oGH959-60) is pGH502. Plasmids and oligonucleotides are listed in *Supplementary files 1B and 1C*, respectively. The identity of the HEK293 cells was 100% matched by STR Profiling at ATCC. Cells were negative for mycoplasma using ATCC PCR kit #30–1012K on January 10, 2017.

## Recombinant AP2 cores

Bicistronic vectors expressing the hexahistidine-tagged mouse AP2 β2 trunk along with mouse μ2 were based on pGH424 (*Hollopeter et al., 2014*) and generated as follows: Mutations were introduced in μ2 using Gibson assembly, and primers oGH847-8 for E302K and oGB24 + 33 for T156A. A thrombin protease site (not utilized in this study) was inserted after amino acid 236 of μ2 using oGB34-35. The construct encoding wild type β2 trunk/μ2 is pGB21, while β2 trunk/μ2(E302K) is pGB19 and β2 trunk/μ2(T156A) is pGB31. The bicistronic vector expressing the GST-tagged mouse AP2 α trunk along with rat σ2 (pGH504) was generated as follows: The α trunk/σ2 hemicomplex was amplified from the original expression vector (*Collins et al., 2002*) using primers oGH1249-50 and recombined with the pACYCDuet vector backbone (amplicon oGH1246-47) using Gibson assembly. The thrombin cleavage site between the α trunk and GST was replaced with the human rhinovirus (HRV) C3 protease site (not utilized in this study) with oGH1227-8 and Gibson assembly. Simultaneous expression of all four subunits of the AP2 core was achieved by transforming *E. coli* BL21

(DE3) cells (New England Biolabs, Ipswich, MA) with a pair of the bicistronic vectors and co-selecting for ampicillin resistance (pGB21, pGB19 or pGB31) and chloramphenicol resistance (pGH504). To generate phosphorylated AP2 cores, the kinase domain of mouse AAK1 (amino acids 1–325, BC141176.1) was amplified from plasmid template using oEP17-18 and inserted into the pRSFDuet backbone amplified with oEP13 + oGH338 using Gibson assembly. The AP2 subunits were co-expressed with this vector (pEP82) by including selection for kanamycin resistance. Plasmids and oligonucleotides are listed in *Supplementary files 1B and 1C*, respectively.

2xYT culture media containing the appropriate antibiotics was inoculated with an overnight culture of bacterial cells expressing AP2 cores and incubated at 37°C with shaking at 180–200 RPM until $OD_{600}$ = ~1.0. The temperature was decreased to 18°C for 1 hr before expression was induced with 100 µM isopropyl β-D-1-thiogalactopyranoside (IPTG) for 20–24 hr. Cells were harvested by centrifugation, washed with lysis buffer (see below), and snap frozen in liquid N2 prior to storage at −80°C. Cell pellets were resuspended in 50 mL lysis buffer per liter of initial culture volume. GST lysis buffer consists of PBS pH 7.4, 1 mM DTT, 60 µg/mL DNase I (grade II from bovine pancreas, Roche), 2.5 mM $MgCl_2$, 0.3 mg/mL lysozyme (Sigma), and 1 mM phenylmethylsulfonyl fluoride (PMSF) (Millipore, Billerica, MA) with the addition of 1 tablet of cOmplete EDTA-free Protease Inhibitor Cocktail (Roche) for every 100 mL. The cell slurry was sonicated (20 s pulses at 20% amplitude for a total of 8 min; 50 mL at a time) using a Q700 sonicator (Qsonica, Newtown, CT). Lysates were cleared by centrifugation (~20000 x g) and filtration (0.2 µm). Cleared filtrate was rotated for 1 hr at 4°C with equilibrated GST resin (GE Healthcare Life Sciences, Uppsala, Sweden), 1 ml resin per liter of the initial culture volume. The filtrate and resin were poured over a gravity column, washed with PBS pH 7.4 + 1 mM DTT, and AP2 was eluted with 50 mM Tris, 150 mM NaCl, 10 mM reduced glutathione, and 1 mM DTT, pH 8.0 (Fischer BioReagents, Fairlawn, NJ). The eluate buffer was exchanged with TBS pH 7.6 + 1 mM DTT and AP2 cores were concentrated to ~0.5 mg/mL using an Amicon Centrifugal Filter Unit with a 100 kDa cutoff (Merck Millipore). Aliquots of AP2 cores were snap frozen in liquid N2 prior to storage at −80°C.

## Recombinant NECAPs

The NECAP used in *Figure 4C* had an N-terminal hexahistidine tag, HaloTag and TEV protease site. Mouse NECAP2 (BK000657.1) was amplified from pGH502 using primers oGH1205-6, and recombined with the backbone of the 6xHis-HaloTag-APA expression vector pGH493 (*Hollopeter et al., 2014*) (amplified with oGH853 + 1204) using Gibson assembly. This replaced the APA domain with NECAP2 and generated pGH503.

Because we sometimes noted heterogeneity in our recombinant NECAPs, we built a set of vectors to purify NECAPs with cleavable C-terminal affinity tags. The NECAPs used for pulldowns in *Figure 4B* and *Figure 6—figure supplement 1B* were purified using the Intein Mediated Purification with an Affinity Chitin-binding Tag (IMPACT, NEB) system, with slight modifications. The *Mxe* GyrA intein tag and chitin binding domain (CBD) from pTXB1 (NEB) were amplified using oGB52-3 (which adds a hexahistidine tag onto the C-terminus of the CBD). This amplicon was inserted into the pET21b expression vector (PCR product of oGB47 + oGH1231) in a three-piece Gibson reaction along with one of the following NECAPs: pGB29 included *C. elegans* NCAP-1 (NM_061997.5, amplified with oGB48-9 from pGH500), pGB27 included *M. musculus* NECAP1 (BK000656.1, amplified with oGB48 + 50 from pGH501) and pGB28 included *M. musculus* NECAP2 (BK000657.1, amplified with oGB48 + 51 from pGH502). The NECAPs were amplified such that they include the N-terminal HaloTag and TEV protease site from the mammalian expression vectors (see Tissue culture pulldowns). To generate the NECAPs used in *Figure 6—figure supplement 1B*, mutations were introduced in NECAP2 (pGB28) using Gibson assembly with oEP642-3 (A32D; pGB91) and oEP644-5 (S87N; pGB94). To generate a HaloTag control protein for these pulldowns (pGB81), pGB28 was amplified with oGB52 + 174 and recombined in a one-piece Gibson reaction to remove the NECAP sequence. All plasmids and oligonucleotides are listed in *Supplementary file 1B and 1C*, respectively.

*E. coli* BL21(DE3) cells (NEB) expressing HaloTag-NECAPs-intein-6xHis (pGB27-9, pGB91, pGB94), 6xHis-HaloTag-NECAP2 (pGH503), the 6xHis-HaloTag control (pGH494) (used in *Figure 4B and C*) (*Hollopeter et al., 2014*), or the HaloTag-intein-6xHis control (pGB81) (used in *Figure 6—figure supplement 1B*) were cultured in Terrific Broth (RPI) containing carbenicillin selection.

Cultures were subsequently grown as described for AP2 cores (above) except expression was induced with 300 μM IPTG and cells were harvested after 16–20 hr incubation at 18°C.

For 6xHis-HaloTag-NECAP2 (pGH503) and 6xHis-HaloTag (pGH494) cell pellets collected from 500 mL of culture were re-suspended in 50 mL nickel lysis buffer (20 mM HEPES pH 7.5, 500 mM NaCl, 30 mM imidazole, 5% glycerol, 5 mM BME) with the addition of 60–200 μg/mL DNase I (grade II from bovine pancreas, Roche), 2.5 mM MgCl$_2$, 1 tablet of cOmplete EDTA-free Protease Inhibitor Cocktail (Roche), 0.3 mg/mL lysozyme (Sigma), and 1 mM PMSF. Cells were lysed by sonication using a Q700 sonicator (Qsonica) at 20% amplitude with 20 s pulses for 4 min total, centrifuged at ~20000 x g for 30 min and filtered to 0.2 μm. The hexahistidine-tagged proteins were purified using nickel-charged 5 mL HiTrap Chelating HP columns (GE Healthcare Life Sciences) on a BioLogic LP system (BioRad). After sample loading, columns were washed with nickel lysis buffer and bound proteins were eluted with nickel elution buffer (nickel lysis buffer with 1 M imidazole). Fractions (1 mL each) containing the protein of interest were combined, buffer exchanged with 20 mM Tris pH 8, 5% glycerol, 1 mM EDTA, 1 mM DTT, and 50 mM NaCl and purified by ion exchange chromatography using a HiTrap Q HP column (GE Healthcare Life Sciences) on the BioLogic LP system (BioRad). After sample loading, proteins were eluted using a salt gradient (50 mM to 1000 mM NaCl). Elution fractions containing the protein of interest were combined and aliquots were snap frozen in liquid N2 prior to storage at −80°C.

To purify recombinant NECAPs (HaloTag-NECAPs-intein-6xHis) or the HaloTag-intein-6xHis control we used a modified version of the IMPACT system. Cell lysis was performed as described for other hexahistidine-tagged proteins (above) except BME was omitted from the lysis buffer to inhibit intein self-cleavage. Cleared lysates were loaded onto gravity columns containing nickel resin (Thermo Scientific, 0.5 mL per 500 mL initial culture volume) and washed extensively. Columns were then flushed with nickel lysis buffer containing 40 mM BME, plugged, and incubated at 4°C for 30 to 40 hr in order to cleave proteins from the intein tag. The released proteins were collected, buffer exchanged with TBS pH 7.6 + 1 mM DTT, and concentrated using an Amicon Centrifugal Filter Unit with 50 kDa MW cutoff (Merck Millipore) or with 30 kD MW cutoff for the HaloTag control. Aliquots were snap frozen in liquid N2 prior to storage at −80°C.

## In vitro pulldown assays

Pulldown assays were performed essentially as described in *Hollopeter et al., 2014* except the protease cleavage step was 3 hr and the input gel sample represented 20% of the 'prey' protein mixture.

## Nerve ring microscopy

Worms expressing APA-2:GFP (*mewSi1*) and RFP:NCAP-1 (*mewSi2*) in an *ncap-1* background were imaged on an Zeiss LSM 880 confocal microscope (Biotechnology Resource Center) with a 40x water immersion objective. APA-2:GFP(*mewSi1*) is molecularly similar to APA-2:GFP(*oxSi254*) (*Gu et al., 2013*) except the entry clones ([4-1]*Pdpy-30*, [1-2]*apa-2* cDNA, and [2-3]*GFP:unc-54 3'UTR*) were recombined with the [4-3] MosSCI vector (pCFJ210) to target the ttTi4348 site in EG6701 worms (*Frøkjær-Jensen et al., 2012*). Fluorophores were excited with 488 nm (GFP) and 561 nm (RFP) lasers. All strains were imaged in one session with the same laser settings. For each worm, a single confocal slice through the approximate sagittal section of the nerve ring was analyzed in Fiji (*Schindelin et al., 2012*). Two regions of interest (ROI) corresponding to both dorsal and ventral sections of the nerve ring along with an ROI outside of the worm (to control for background signal) were user defined. The average pixel intensities in both the GFP and RFP channels were determined and the background values were subtracted from the nerve ring values.

## CRISPR-Cas9 generation of μ2 and NCAP-1 mutations

CRISPR/Cas9 edits were generated using the *dpy-10* co-conversion strategy (*Arribere et al., 2014*) with ribonucleoprotein (RNP) complexes (*Paix et al., 2015*). For generating μ2 mutations, gonads of young adult hermaphrodites were injected with RNP mixes containing ~3.7 μg/μL Cas9 (purified in-house), 1 μg/μL tracrRNA, 0.08 μg/μL *dpy-10* crRNA, 16.7 μM *dpy-10(cn64)* roller repair, 0.4 μg/μL gene-specific crRNA, and 16.7 μM gene-specific repair. The gene-specific crRNAs and oligonucleotide repairs were as follows: to generate μ2(E306K) – rEP360 and oGB154 (introduces an *Nru*I site

for genotyping), to generate μ2(T160A) – rGB156 and oGB130 (introduces an *Ngo*MIV site), and to generate μ2(R440S) – rGB155 and oGB159 (introduces a *Pvu*I site). F1 progeny exhibiting the roller phenotype were placed on individual culture plates and allowed to produce offspring for 1–2 days before being lysed in 50 μL 1X Phusion GC buffer (NEB) with 0.4 U Proteinase K (NEB) by freezing at −80°C and then heating at 65°C for 1 hr followed by 95°C for 15 min. The targeted region of the genome was amplified using PCR and the amplicon was digested to identify correctly edited worms. Mutations were confirmed by sequencing the PCR product.

To generate NCAP-1 missense mutations, injection mixes were prepared with 5 μM of each repair. The gene-specific crRNAs and oligonucleotide repairs were as follows: to generate RFP: NCAP-1(A29D) in the microscopy strain GUN62 – rEP254 and oEP674, to generate RFP:NCAP-1 (A29D) in the TEV assay strain GUN106 – rEP254 and oEP680, and to generate RFP:NCAP-1(S84N) – rEP676 + 700 (each at 0.2 μg/μL) and oEP701 (introduces an *Xba*I site).

All materials and resources described in this article are available upon request with no restrictions.

# Acknowledgements

We thank Erik Jorgensen, Alejandro Sánchez Alvarado, and the Stowers Institute for Medical Research for support. Ho Yi Mak provided lab space, reagents, and technical advice during the early stages of this work at the Stowers Institute. Technical assistance at the Stowers Institute was provided by Jim Vallandingham (Bioinformatics), Kendra Walton (Molecular Biology), and Valerie Neubauer (Tissue Culture). We thank the labs of Maurine Linder, Joshua Chappie, Carolyn Sevier, Hector Aguilar-Carreno, Carrie Adler, and Scott Emr for technical advice, reagents and the use of equipment. We thank Anthony Bretscher, Carolyn Sevier, Chris Fromme, Joshua Chappie, Carrie Adler, and Holger Sondermann for constructive criticism that greatly improved the manuscript. The Zeiss LSM 880 confocal microscope was purchased with Cornell University Biotechnology Resource Center instrument grants NYSTEM CO29155 and NIH S10OD018516. GMB and EAP were supported by an NIH training grant GM007273-43. GMB is supported by an NSF graduate research fellowship DGE-1650441.

# Additional information

## Funding

| Funder | Grant reference number | Author |
|---|---|---|
| National Science Foundation | Graduate Research Fellowship DGE-1650441 | Gwendolyn M Beacham |
| National Institutes of Health | Training Grant GM007273-43 | Gwendolyn M Beacham Edward A Partlow |

The funders had no role in study design, data collection and interpretation, or the decision to submit the work for publication.

## Author contributions

Gwendolyn M Beacham, Conceptualization, Formal analysis, Funding acquisition, Validation, Investigation, Visualization, Methodology, Writing—original draft, Writing—review and editing; Edward A Partlow, Conceptualization, Resources, Investigation, Visualization, Methodology, Writing—review and editing; Jeffrey J Lange, Conceptualization, Data curation, Software, Formal analysis, Methodology, Writing—review and editing; Gunther Hollopeter, Conceptualization, Resources, Supervision, Funding acquisition, Validation, Investigation, Visualization, Methodology, Project administration, Writing—review and editing

## Author ORCIDs

Gunther Hollopeter http://orcid.org/0000-0002-6409-0530

Decision letter and Author response
Decision letter https://doi.org/10.7554/eLife.32242.015
Author response https://doi.org/10.7554/eLife.32242.016

## Additional files

### Supplementary files

• Supplementary file 1. (A) Strains (B) Plasmids (C) Oligonucleotides.

DOI: https://doi.org/10.7554/eLife.32242.011

• Transparent reporting form

DOI: https://doi.org/10.7554/eLife.32242.012

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
