## [Decision Letter]

Thank you for submitting your article "NECAPs are negative regulators of the AP2 clathrin adaptor complex" for consideration by *eLife*. Your article has been favorably evaluated by Vivek Malhotra (Senior Editor) and three reviewers, one of whom, Suzanne Pfeffer, is a member of our Board of Reviewing Editors. The reviewers have discussed the reviews with one another and the Reviewing Editor has drafted this decision to help you prepare a revised submission.

NECAP1 has been shown by McPherson and colleagues (2013) to regulate AP-2:clathrin interactions, thereby controlling endocytic vesicle size, number and cargo during endocytosis in fibroblasts and cultured hippocampal neurons. It does this by modulating accessory protein binding to AP-2 in coordination with clathrin binding. Here, the authors identify ncap-1 as a suppressor of fcho-1 mutants in *C. elegans*. Loss of NCAP-1 restores AP2 activity in fcho-1 mutants; Ce and murine NCAP-1 or 2 can rescue. The authors show that NECAPs bind open and phosphorylated AP2 core and seem to stabilize the phosphoform in vivo, thereby regulating its activity. These genetic data complement those previously reported from the McPherson group and extend our understanding of how various NECAP complexes may regulate clathrin function.

Overall, the reviewers thought the data were high quality and suggest the following experiments to bring the story to the level of impact that will make it suitable for presentation in *eLife*.

1) Do the nature of some of the NECAP point mutants isolated provide any insight into a mechanism by which loss of AP2 regulatory function may occur? For example, do the A29D or S84N prevent interaction between NECAP and AP-2 (either the open conformation or the phosphorylated complex)? Or do these mutations simply affect stability of NECAP? Investigating this idea further may reveal key insights into the specific role of NECAP proteins in modulating AP2 activity.

2) Please include some quantification of the data shown in Figure 4.

3) The model presented seems rather oversimplified. How precisely do the authors envision that loss of NECAP bypasses FCHO? In FCHO mutant animals, AP2 remains largely closed and hypo-phosphorylated, based on the authors' previous work. Since NECAP proteins fail to bind AP2 under these conditions, it is unclear how bypass is achieved in the absence of FCHO. The authors do not clarify this issue in the Discussion, but this should really be directly addressed.

4) In Figure 5, the authors analyze the effects of AP-2 activating mutations on relative enrichment of NECAP and AP-2 in nerve rings. This follows data in Figure 4 that, in vitro, NECAP binds poorly to the closed form of AP-2 core. Considering the in vitro data, a reader might expect that the ratio of NECAP to AP-2 in fcho1 mutants would be lower than wild-type since AP-2 in fcho1 animals is more closed than wild-type. Consequently, in describing the results in the subsection “NCAP-1 localizes with constitutively open AP2 in vivo”, the authors might want to explain why the ratios in wild-type and fcho1 mutants are the same (perhaps because of NECAP binding to AP-2 appendages?).

5) The Ritter papers should be more explicitly described in the Introduction. Also, if there was any way to link the papers – i.e., incorporate the previous finding into a new Figure 6 model, it would help the non-Fcho experts understand this solid advance.

6) Figure legends should be clarified so that they can be read independent of the manuscript text. Figure 6 legend is very unclear for non-experts. For example, Fig, 2E phosphorylation assay [of what?].

[Editors' note: further revisions were requested prior to acceptance, as described below.]

Thank you for resubmitting your work entitled "NECAPs are negative regulators of the AP2 clathrin adaptor complex" for further consideration at *eLife*. Your revised article has been favorably evaluated by Vivek Malhotra (Senior Editor), a Reviewing Editor, and two reviewers.

The reviewers felt you have done a very good job addressing their concerns. One reviewer noted that just for clarity, you might want to address the apparent discrepancy between the in vitro binding of mouse NECAP2 to E302K AP-2 cores in Figure 6 and the absence of recruitment of *C. elegans* NCAP-1 to nerve rings in the corresponding E306K AP-2 worm mutant. Perhaps at the end of the subsection” Missense mutations render NECAPs functionally inactive”, you could include something like, "In contrast to these in vitro results, the absence of NCAP-1 recruitment to nerve rings in the E302K *C. elegans* mutant is presumably due to phosphorylation of this mutant in vivo".

---

## [Author Response]

[…] Overall, the reviewers thought the data were high quality and suggest the following experiments to bring the story to the level of impact that will make it suitable for presentation in eLife.1) Do the nature of some of the NECAP point mutants isolated provide any insight into a mechanism by which loss of AP2 regulatory function may occur? For example, do the A29D or S84N prevent interaction between NECAP and AP-2 (either the open conformation or the phosphorylated complex)? Or do these mutations simply affect stability of NECAP? Investigating this idea further may reveal key insights into the specific role of NECAP proteins in modulating AP2 activity.

The reviewers astutely recognized that further analysis of the special missense mutations in NCAP-1 isolated as bypass suppressors of fcho-1 mutants would yield mechanistic insight into how NECAPs negatively regulate AP2. We employed a multipronged approach to these experiments. First, we engineered the A29D and S84N mutations into worms expressing a single copy RFP:NCAP-1 transgene using CRISPR. We then evaluated the effect of these mutations on NCAP-1 stability and activity using our in vivo assays. Even though these mutant forms of NCAP-1 are stable according to western blot analysis, they fail to complement *ncap-1* mutants or restore the closed form of AP2 in *fcho-1* mutants, and do not associate with open, hyper-phosphorylated AP2 in vivo. Because these residues are highly conserved, we also introduced the equivalent mutations into vertebrate NECAP2 proteins and verified that they produced stable recombinant proteins. Interestingly, when these mutated NECAP2 proteins were used as bait in in vitro pulldown assays, they appeared to have selectively lost the ability to bind to the phosphorylated AP2 core. We interpret these results as further support of our model that the conserved N-terminal region of NECAPs known as the PHear domain negatively regulates the AP2 complex by directly binding the phosphorylated core. The data have been assembled into a new Figure 6 along with the Figure 6—figure supplement 1 and described in the Results and Discussion.

Results: “In the *fcho-1* suppressor screen, we isolated two independent missense mutations in the N-terminal PHear domain of NCAP-1: A29D and S84N (Figure 1). […] These data indicate that the missense mutations in NCAP-1 obstruct AP2 inactivation by preventing NCAP-1 from binding to the phosphorylated AP2 core.”

Discussion: “Based on the missense mutations that disrupt NECAP activity in vivo and AP2 binding in vitro (Figure 6 and Figure 6—figure supplement 1), we believe this inhibitory function may involve a direct interaction of the N-terminal PHear domain (Ritter et al., 2007) with the phosphorylated AP2 core.”

We also evaluated a third missense NCAP-1 mutation (E108V) using the same approaches and determined that this mutant exhibited wild-type activity in our assays. One obvious explanation for this discrepancy is that there are additional mutations in the strain we isolated from our mutagenesis screen that are partially or entirely responsible for the suppression of *fcho-1*. Thus, we have decided to remove this confusing NCAP-1 mutant from the list of alleles that suppress fcho-1 until we further delineate the mutations responsible for the suppression we observe in the original strain.

2) Please include some quantification of the data shown in Figure 4.The data in Figure 4 represents stained protein gels or western blot analysis of AP2 complexes isolated from pulldown assays using NECAP proteins as the bait. We have quantified the intensity of the AP2 α subunit in each pulldown, normalized to a negative control (HaloTag alone) and included these values in Figure 4. We also performed this analysis for the pulldowns performed in response to Point 1 (Figure 6—figure supplement 1).3) The model presented seems rather oversimplified. How precisely do the authors envision that loss of NECAP bypasses FCHO? In FCHO mutant animals, AP2 remains largely closed and hypo-phosphorylated, based on the authors' previous work. Since NECAP proteins fail to bind AP2 under these conditions, it is unclear how bypass is achieved in the absence of FCHO. The authors do not clarify this issue in the Discussion, but this should really be directly addressed.

We appreciate this feedback and recognize that this point needed clarification in the Discussion. Essentially, we propose that in the absence of NCAP-1, the lifetime of the open state is increased. We hypothesize that without allosteric activation by FCHO-1, AP2 still attains the open, phosphorylated state, albeit at a reduced rate. In this context (*fcho-1* mutants), NCAP-1 inactivates these AP2 complexes, causing the closed, inactive form of AP2 to predominate in *fcho-1* mutants. Removal of NCAP-1 tips the balance toward maintaining the open, active AP2 conformation. The *fcho-1* suppressor screen appears to have yielded three potential routes to escape inactivation by NCAP-1 that enable AP2 to accumulate in more active, if unregulated forms: 1) open AP2 mutations that constitutively bind NCAP-1 but resist its action, 2) phosphorylation site mutations in AP2 that evade NCAP-1 binding and 3) mutations in the *ncap-1* gene itself that produce a defective protein. We have clarified our model in the Discussion and in our new model figure (Figure 7).

Discussion: “…mutations that directly disrupt NCAP-1 (class 3) probably enable a basal level of AP2 activity to be sustained by complexes that succeed in attaining the open, phosphorylated state in lieu of allosteric activation by muniscins.”

Figure 7 legend: “In the absence of muniscins, AP2 activation is greatly reduced. The *fcho-1* suppressor screen isolated three classes of mutations that enable AP2 to remain active in lieu of muniscins…”

4) In Figure 5, the authors analyze the effects of AP-2 activating mutations on relative enrichment of NECAP and AP-2 in nerve rings. This follows data in Figure 4 that, in vitro, NECAP binds poorly to the closed form of AP-2 core. Considering the in vitro data, a reader might expect that the ratio of NECAP to AP-2 in fcho1 mutants would be lower than wild-type since AP-2 in fcho1 animals is more closed than wild-type. Consequently, in describing the results in the subsection “NCAP-1 localizes with constitutively open AP2 in vivo”, the authors might want to explain why the ratios in wild-type and fcho1 mutants are the same (perhaps because of NECAP binding to AP-2 appendages?).

We agree that the lack of difference between wild type and fcho-1 mutants is confusing at first glance. We interpret this as support of the model that worm NECAP (NCAP-1) is not usually coincident with AP2 on the membrane. In data not included in this manuscript, we failed to observe co-localization of RFP:NCAP-1 with AP2 on the plasma membrane of coelomocytes. Instead, we believe that open AP2 mutants *artificially* recruit NCAP-1 to sites of endocytosis and that this effect is most striking at the nerve ring, a tissue rich in membrane and synapses. However, we have not observed a similar enrichment on the membrane of coelomocytes. The limited resolution of our microscopy, coupled with the small diameter of the neuronal processes in the nerve ring, does not allow us to discern whether NCAP-1 and AP2 are truly co-localized on the membrane. We have included statements regarding our interpretation of the microscopy data in the Results and Discussion.

Results: “In both wild type and *fcho-1* mutant worms, NCAP-1 was not overtly enriched compared to AP2. The low level of NCAP-1 relative to AP2 may indicate that in worms, NCAP-1 is not stably associated with AP2 on membranes.”

Results: “In other words, NCAP-1 was abnormally recruited to the nerve ring in the open AP2 mutant…”

Discussion: “Open, hyper-phosphorylated AP2 mutants (class 1) appear to recruit NECAPs in excess…”

It is worth noting that NCAP-1 lacks the consensus C-terminal α appendage-binding motif (WXXF) and instead ends with LLDF, so we are not confident NCAP-1 binds the α appendage of AP2 in *C. elegans*. Nevertheless, it is possible the low levels of RFP:NCAP-1 we observe in worms with and without *fcho-1* may represent NCAP-1 bound to the AP2 appendages and we have included this as a possible explanation in the Results.

Results: “It is possible that the basal level of localization observed in both wild type and *fcho-1* mutants may represent NCAP-1 bound to the AP2 appendages (Ritter et al., 2013; Ritter et al., 2003).”

5) The Ritter papers should be more explicitly described in the Introduction. Also, if there was any way to link the papers – i.e., incorporate the previous finding into a new Figure 6 model, it would help the non-Fcho experts understand this solid advance.

We have expanded the summary of previous work on NECAPs in the introduction to include more of the discoveries of Ritter and colleagues.

Introduction: “NECAPs were originally identified as endocytic accessory proteins through proteomic analysis of clathrin-coated vesicles (Wasiak et al., 2002) and were shown to bind the AP2 α appendage via a C-terminal WXXF motif (Ritter et al., 2004; Ritter et al., 2003). […] By contrast, ubiquitously-expressed NECAP2 has been proposed to recruit a different clathrin adaptor, AP1, to early endosomes to facilitate fast recycling of receptors back to the cell surface (Chamberland, Antonow, Dias Santos, and Ritter, 2016).”

We have constructed a new model figure (now Figure 7, see legend below) that synthesizes the previous work and incorporates our new findings using language we hope will be more accessible to readers unfamiliar with FCHo proteins. We have also linked the results in this current manuscript with the previous findings in the Discussion.

Discussion: “We now recognize that these *fcho-1* suppressors occur in three distinct classes: dominant missense mutations that disrupt the closed conformation of AP2 (class 1) or mutate the phosphorylation site on µ2 (class 2), and recessive mutations that inactivate NCAP-1 (class 3) (Figure 7). […] The wealth of mutations in AP2 and NECAP identified in the *fcho-1* suppressor screen, combined with our biochemical and imaging data, reveal discrete steps of AP2 conformational and phosphorylation changes during its recycling (Figure 7).”

Discussion: “Despite this proposed divergence, both vertebrate NECAPs rescue loss of NCAP-1 in the nematode, as does a fungal NECAP that lacks the C-terminal α adaptin-ear binding WXXF motif. These results indicate that the capacity to negatively regulate AP2 has been conserved among NECAPs.”

6) Figure legends should be clarified so that they can be read independent of the manuscript text. Figure 6 legend is very unclear for non-experts. For example, Fig, 2E phosphorylation assay [of what?].

We thank the reviewers for helping us to improve the clarity of our manuscript and have modified the figure legends to be clearer. The following two examples specifically address the reviewers’ suggestions:

Figure 7 (formerly Figure 6): “Muniscins allosterically activate AP2 to form a stable association with the membrane. […] Each class disrupts the AP2 inactivation pathway and promotes accumulation of AP2 at discrete steps in the cycle (gray arrows).”

Figure 2: We specified that both the phosphorylation and protease sensitivity assays are of “μ2”.

[Editors' note: further revisions were requested prior to acceptance, as described below.]

The reviewers felt you have done a very good job addressing their concerns. One reviewer noted that just for clarity, you might want to address the apparent discrepancy between the in vitro binding of mouse NECAP2 to E302K AP-2 cores in Figure 6 and the absence of recruitment of C. elegans NCAP-1 to nerve rings in the corresponding E306K AP-2 worm mutant. Perhaps at the end of the subsection” Missense mutations render NECAPs functionally inactive”, you could include something like, "In contrast to these in vitro results, the absence of NCAP-1 recruitment to nerve rings in the E302K C. elegans mutant is presumably due to phosphorylation of this mutant in vivo".

We recognize that our summary of the results was unclear and insufficient. We have added a sentence of clarification at the end of the subsection “Missense mutations render NECAPs functionally inactive”:

Results – “The in vitro pulldown assays suggest that the primary reason the NCAP-1 missense mutants are inefficiently recruited to the nerve ring by the open AP2 mutation in vivo may be that the corresponding adaptor complexes are also phosphorylated.”

We also address the discrepancy in the Discussion:

Discussion – “However, an interaction with these constitutively open, non-phosphorylated cores in vitro may not be relevant; functionally inactive NECAPs (the A29D and S84N mutants) appear to bind these cores in vitro(Figure 6—figure supplement 1), but do not associate with the equivalent, albeit hyper-phosphorylated, open AP2 mutation in vivo (Figure 6—figure supplement 1). The non-phosphorylated open AP2 core may not be a physiological state of AP2 and might not accurately simulate the in vivo interaction between NECAP and AP2.”